# Elytra: A Flexible Framework for Securing Large Vision Systems

## Abstract

Adversarial attacks have emerged as a critical threat to autonomous driving systems. These attacks exploit the underlying neural network, allowing small – almost invisible – perturbations to alter the behavior of such systems in potentially malicious ways, *e.g.*, causing a traffic sign classification network to misclassify a stop sign as a speed limit sign. Prior work in hardening such systems against adversarial attacks has looked at fine-tuning of the system or adding additional pre-processing steps to the input pipeline. Such solutions either have a hard time generalizing, require knowledge of adversarial attacks during training, or are computationally undesirable. Instead, we propose a framework called ELYTRA to take insights for parameter-efficient fine-tuning and use low-rank adaptation (LoRA) to train a lightweight security patch (or patches), enabling us to dynamically patch large pre-existing vision systems as new vulnerabilities are discovered. We demonstrate that the ELYTRA framework can patch pre-trained large vision models to improve classification accuracy by up to 24.09% in the presence of adversarial examples.

## 1 Introduction

Large-scale vision models, such as *vision transformers* (ViT) (Dosovitskiy et al., 2021; Liu et al., 2021b), have revolutionized the capabilities of autonomous driving systems (Liu et al., 2021a; Bai et al., 2023; Saha et al., 2022; Li et al., 2022; Yang & Liu, 2021). However, such models are *inherently* vulnerable to adversarial attacks (Szegedy et al., 2014), *i.e.*, attacks that perform small input modifications – often imperceptible to the naked eye – could cause the model to perform in a *malicious* manner, *e.g.*, classifying a stop sign as a speed limit sign. With the increasing deployment of these models within *critical* autonomous driving systems, the possibility of adversarial attacks poses a serious threat to the safety and reliability of such systems, in particular, for tasks such as the classification and detection of traffic signs (Eykholt et al., 2018; Wei et al., 2022; Zhong et al., 2022; Hsiao et al., 2024; Lu et al., 2017; Aung et al., 2017; Etim & Szefer, 2024b; Sitawarin et al., 2018).

As advanced vision systems continue to grow and become more complicated (Liang et al., 2022; Liu et al., 2023; Zhang et al., 2022), traditional mitigation techniques either become computationally undesirable, such as adversarial training (Tramèr et al., 2018; Wan et al., 2020; Etim & Szefer, 2024a;b; Dehghani et al., 2023), or fail to generalize (Etim & Szefer, 2024b), such as input pre-processing (Qiu & Qiu, 2020). Adversarial fine-tuning is standard for making models robust against such attacks; however, retraining models against new attacks to increase the breadth of protection is both time-intensive and costly. In particular, autonomous systems must be robust against adversarial inputs to maintain and provide safe operation; thus, being unable to defend against new attacks leaves these systems vulnerable and potentially hazardous in lieu of adversarial entities.

Rather than focusing on retraining and fine-tuning the underlying vision system or modifying the input images via additional pre-processing, we want to explore whether we can simply roll out a *security adapter* for our network like in other traditional computing systems. *I.e.*, when a *new* vulnerability is discovered, can we simply train a security adapter for the network rather than retraining the *entire* network? For this system to make any sense, we need an *efficient* way to fine-tune large models. Fortunately, recent advances

Figure 1: Overview of ELYTRA. **(a)** Diverse adversarial attacks cause misclassifications in large vision models. **(b)** ELYTRA trains a lightweight LoRA adapter (security patch) per attack type while keeping all base model weights frozen. **(c)** Adapters compose additively into a single multi-threat defense.

in efficient parameter fine-tuning have provided methods to address this problem, in particular, *low-rank adaptation* (LoRA) (Hu et al., 2022). We summarize our *key insight* below as:

> We can use LoRA as a lightweight framework for security adapters.

In this work, we propose ELYTRA, a lightweight framework for patching large pre-trained vision systems *post hoc*, making use of LoRA as the mechanism against security vulnerabilities. We present an overview of the ELYTRA framework in Figure 1. Our main contributions are as follows.

- We demonstrate that Elytra succeeds both in being lightweight (99.7% reduction in trainable parameters and up to 15× faster than retraining) and in hardening the network against different vulnerabilities (improving classification accuracy by up to 24.09%).

- By rolling out these security patches sequentially, we overcome catastrophic forgetting and concept loss, enabling the composition of multiple patches without needing more complex and computationally demanding techniques.

- We validate our framework on multiple vision transformer architectures using a large-scale dataset of traffic signs and successfully harden these systems to potent adversarial attacks.

## 2 Preliminaries

### 2.1 Securing large vision models

Zhao et al. (2025) highlights the complications related to the high-volume training of machine learning models due to the risks involving data poisoning from sources, including, but not limited to, adversarial perturbations passed as bona fide data. The hardening of image classification systems to defend against such attacks is a well-studied area (Madry et al., 2019; Chakraborty et al., 2021) and vital for systems that involve safety on a large scale, *i.e.*, traffic sign classification (Aung et al., 2017; Etim & Szefer, 2024a;b; Pavlitska et al., 2023; Sitawarin et al., 2018). Pavlitska et al. (2023) conducted a comprehensive survey on adversarial attacks that have been investigated in the scope of traffic sign classification and found that these attacks included digital attacks, synthetic real-world attacks, and physical perturbations. Aung et al. (2017) found that during training, combining defensive distillation and adversarial training, the model was able to improve generalization, making the model less sensitive to adversarial perturbations.

**Vision transformers.** Vision transformers (ViTs) have been gaining popularity for a wide range of autonomous navigation tasks. Usage can be seen in Ando et al. (2023) where they used the vision transformer in a non-linear task involving 3-D semantic segmentation. To facilitate this, they combined a nonlinear layer

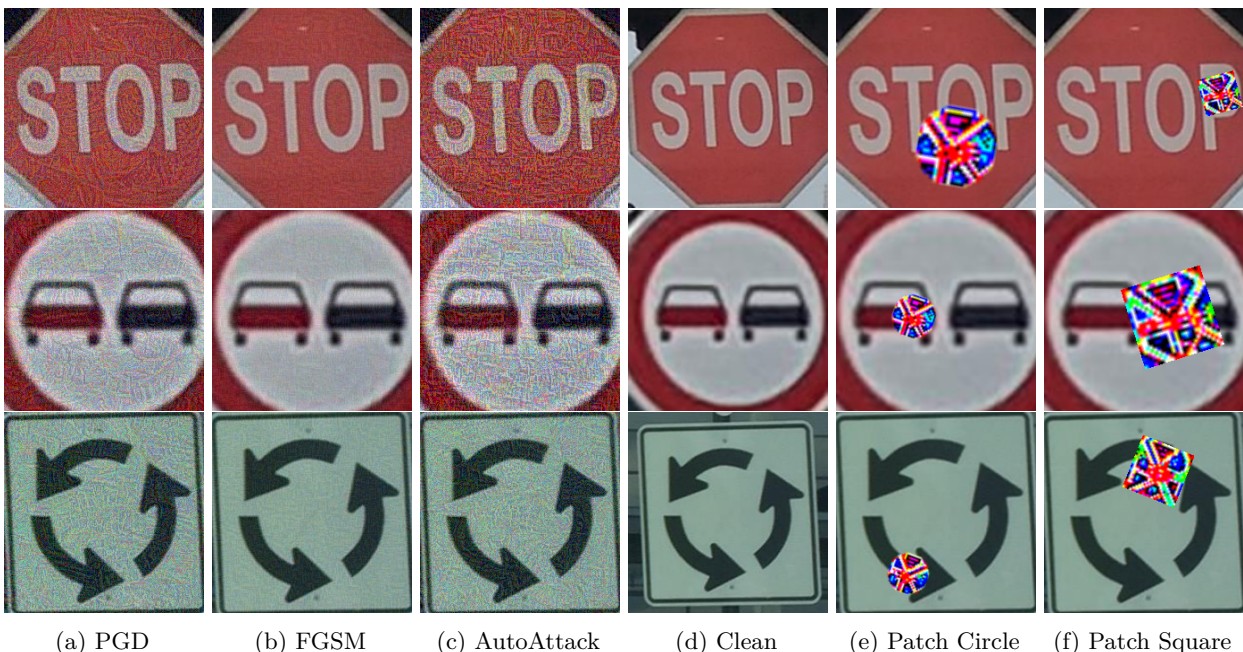

| (a) PGD | (b) FGSM | (c) AutoAttack | (d) Clean | (e) Patch Circle | (f) Patch Square |

Figure 2: An illustration of adversarial perturbations applied to an example set of signs ("Clean"). *N.B.*, that the Euclidean projection onto the feasibility set $\mathcal{S}$ in PGD keeps the distortions minimal at similar step sizes $\varepsilon$ when compared to FGSM. In the first row, we have a "Stop Sign", followed by "No Overtaking", and lastly "Roundabout".

with a pooling layer to reduce the dimensions back to what is expected of the ViT. While experiencing minor issues with their work, they successfully used a pre-trained vision transformer for LiDAR segmentation and classification for potential hazards while using autonomous navigation. Lai-Dang (2024) conducted a survey that highlights the strength of ViTs over typical neural networks in autonomous navigation and detection tasks such as 3-D object detection, lane segmentation, map generation, and leverage self-attention mechanisms to increase complex driving scenarios. In addition to tasks in 3-D space, recent research has shown comparable or better results in traffic sign detection and classification using vision transformers over typical neural networks using various implementations and designs (Kaleybar et al., 2023; Farzipour et al., 2023).

**Security of autonomous systems.** Adversarial attacks against the aforementioned classification systems have been successfully deployed in many different forms and, at their core, are huge security risks for autonomous navigation systems, from the use of stickers (Eykholt et al., 2018; Wei et al., 2022; Hsiao et al., 2024) to the use of shadows (Zhong et al., 2022; Etim & Szefer, 2024b), and even the manipulation of the entire surface of the sign (Lu et al., 2017). Recent studies have started investigating the effectiveness of attacks that leverage natural artifacts to cause misclassifications. In Etim & Szefer (2024a), different types of digital leaves were placed on a sign using a grid search placement method to find the optimal location of placement. These, often subtle, modifications can significantly affect the classification of traffic signs, compromising security in autonomous navigation systems. Goodfellow et al. (2014) finds adversarial attacks, in particular Fast Gradient Signed Method (FGSM) and Projected Gradient Descent (PGD) attacks, to be extremely effective due to the linear nature of image classification systems and the way models approach learning when weight vectors between target classes are similar between different models trained on the same task. Recent work by Savostianova et al. (2024) explored the development of PGD attacks with LoRA applied to the perturbation instead of the parameters, unlike our work, which focused on applying LoRA to the parameters.

## 2.2 Low-rank adaptation

*Low-rank adaptation*, or LoRA (Hu et al., 2022), is an immensely popular technique for *parameter-efficient fine-tuning* (PEFT). This technique is widely deployed for the purpose of adapting large pre-trained models to downstream tasks, *e.g.*, teaching new concepts to text-to-image models (Yeh et al., 2023), fine-tuning large language models (LLMs) (Hu et al., 2022), and targeted peptide design (Park et al., 2025). The core concept of LoRA is actually surprisingly simple. Suppose that we have a weight matrix $\boldsymbol{W} \in \mathbb{R}^{u \times v}$ so that we can define the linear layer as the map $\boldsymbol{x} \mapsto \boldsymbol{W}\boldsymbol{x}$ with $\boldsymbol{x} \in \mathbb{R}^u$. Our goal is to update this weight matrix $\boldsymbol{W}$ in an *efficient* manner. Suppose that we have an update matrix $\Delta \boldsymbol{W}$ which can be decomposed into two low-rank matrices, such that $\Delta \boldsymbol{W} = \boldsymbol{A}\boldsymbol{B}$ with $\boldsymbol{A} \in \mathbb{R}^{u \times r}$ and $\boldsymbol{B} \in \mathbb{R}^{r \times v}$ where $r \ll u$ and $r \ll v$. We then have a parameter-efficient technique for updating weight matrices. *N.B.*, this update matrix (or collection of matrices for full neural networks) $\Delta \boldsymbol{W}$ is referred to as a LoRA matrix or, more simply, as a LoRA.

**Composability.** Typically, each individual LoRA learns a *unique* concept. Suppose that we have a collection of LoRAs $\{\boldsymbol{L}_i\}_{i=1}^n$, $\boldsymbol{L}_i = \boldsymbol{A}_i\boldsymbol{B}_i$ and $(\boldsymbol{A}_i, \boldsymbol{B}_i) \in \mathbb{R}^{u \times r} \times \mathbb{R}^{r \times v}$, applied to some weight matrix $\boldsymbol{W} \in \mathbb{R}^{u \times v}$, naïvely composing these via

$$\boldsymbol{x} \mapsto \boldsymbol{W}\boldsymbol{x} + \left(\sum_{i=1}^n \pi_i \boldsymbol{L}_i\right)\boldsymbol{x}, \tag{1}$$

where $\sum_i \pi_i = 1$ is a set of scalar mixing weights that may result in the destruction of the original concepts learned in each $\boldsymbol{L}_i$. Various methods have been proposed to alleviate this problem, (Liang & Li, 2024; Po et al., 2024; Gu et al., 2023; Zhong et al., 2024) which generally fall into some category of learning to orthogonalize each LoRA $\boldsymbol{L}_i$ to one another or a component, for example, all the $\boldsymbol{B}_i$ matrices in low-rank decomposition (Liang & Li, 2024).

## 3 Proposed framework

In this section, we outline the *threat model*, *i.e.*, the precise definition of attacks against which we aim to harden our vision systems, and then our proposed strategy to achieve this aim. Without loss of generality, we consider a standard computer vision pipeline for classification. We assume the existence of some underlying data distribution $(\boldsymbol{X}, Y) \sim \mathbb{P}_{\text{data}}$ where $\boldsymbol{X}$ is a $\mathbb{R}^d$-valued random variable that denotes the image sample and $Y$ is a $[k]$-valued random variable that denotes the class label with $k$ classes with $d > 0$.[1] We then assume that there exists a neural network $\boldsymbol{f_\theta} \in \mathcal{C}^\alpha(\mathbb{R}^d; \mathbb{R}^k)$[2] parameterized by $\boldsymbol{\theta} \in \mathbb{R}^p$ which maps the images to a probabilistic class vector with $r > 1$ and $p > 0$. Next we assume the existence of some suitable loss function $\mathcal{L} \in \mathcal{C}^\alpha(\mathbb{R}^p \times \mathbb{R}^d \times \mathbb{N})$, *e.g.*, the cross entropy between the predicted and true class labels. The model is then trained to minimize expectation $\mathbb{E}_{(\boldsymbol{X},Y)\sim\mathbb{P}_{\text{data}}}[\mathcal{L}(\boldsymbol{\theta}, \boldsymbol{X}, Y)]$.

Unfortunately, this common type of training, while tremendously successful in training highly useful classification systems, is not robust to adversarially crafted examples (Goodfellow et al., 2014; Moosavi-Dezfooli et al., 2016).

### 3.1 LoRA-based mitigation for a single adversary

We begin by formalizing the adversarial threat model for a single adversary, after which we introduce our LoRA-based mitigation strategy that hardens a model against such attacks.

#### 3.1.1 Threat model

In particular, we consider the attack vector of *injection attacks* – attacks that are inserted into the digital pipeline – which allows the use of *adversarial attacks* (Szegedy et al., 2014) against the targeted vision system. Adversarial attacks have been successfully used to attack a large variety of ML systems (Blasingame & Liu, 2024a;b; Chakraborty et al., 2021; Sitawarin et al., 2018). We make use of the threat model from Madry

---

[1]We let $[k]$ denote the set $\{n \in \mathbb{N} : n \leq k\}$.
[2]We let $\mathcal{C}^\alpha(X; Y)$ denote the class of $\alpha$-th differentiable continuous functions from $X$ to $Y$. If $Y$ is omitted, then $Y = \mathbb{R}$.

et al. (2018), which introduces a set of valid perturbations $\mathcal{S} \subseteq \mathbb{R}^d$ for each data sample $\boldsymbol{X}$, which formalizes the power of the adversary. For image classification $\mathcal{S}$ is chosen to capture the notion of perceptual similarity between images—this could be something as simple as the $\ell^{\infty}$-ball centered at $\boldsymbol{X}$, see Goodfellow et al. (2014). We can thus model the *interplay* between our adversary and the defensive efforts as the following minimax game.

$$\min_{\boldsymbol{\theta}} \mathbb{E}_{(\boldsymbol{X},Y)\sim\mathbb{P}_{\text{data}}} \left[ \max_{\delta\in\mathcal{S}} \mathcal{L}(\boldsymbol{\theta}, \boldsymbol{X} + \delta, Y) \right]. \tag{2}$$

### 3.1.2 Mitigation strategy

As mentioned above, our key observation is that we can use LoRA to train lightweight security adapters, Elytras; thus, we update the minimax game from Equation (2) to only fine-tune using LoRA. However, we should note a *crucial* difference: within this game, we assume that Equation (2) has been played *without* updating $\boldsymbol{\theta}$, *i.e.*, the adversary has had their turn, and now we have *ours*. We define $\boldsymbol{\theta}$ more clearly to express this concept formally. Let $\boldsymbol{\theta}$ denote the ordered collection of parameter matrices $\boldsymbol{\theta} = (\boldsymbol{\theta}_1, \ldots, \boldsymbol{\theta}_n)$ with $\boldsymbol{\theta}_i \in \mathbb{R}^{u_i \times v_i}$ and the Cartesian product of such vector spaces is denoted $\boldsymbol{\theta}$, which is isomorphic to $\mathbb{R}^p$. Next, let $\Delta\boldsymbol{\theta}$ denote the ordered collection of *new* information added to the parameter matrices, $\Delta\boldsymbol{\theta} = (\Delta\boldsymbol{\theta}_j)_{j\in J}$ with index set $J = \{j_h \in [n] : 1 \le h \le m\}$ where $m \le n$ and $j_{h-1} < j_h$; and where the update matrices are defined as $\Delta\boldsymbol{\theta}_j \in \mathbb{R}^{u_j \times v_j}$. *N.B.*, this construction enables us to update only a *subset* of model parameters with $J \subseteq [n]$. Thus, the updated objective of Equation (2) is as follows.

$$\min_{\Delta\boldsymbol{\theta}\in\mathfrak{L}_r} \mathbb{E}_{(\boldsymbol{X},Y)\sim\mathbb{P}_{\text{data}}} \left[ \mathcal{L}\left( \boldsymbol{\theta} + \Delta\boldsymbol{\theta}, \boldsymbol{X} + \arg\max_{\delta\in\mathcal{S}} \mathcal{L}(\boldsymbol{\theta}, \boldsymbol{X} + \delta, Y), Y \right) \right], \tag{3}$$

where blue denotes frozen parameters and orange denotes parameters that are learned (see Figure 1), the operator $+$ denotes the addition between the appropriate model parameters and the update matrices, *i.e.*, we overload the operator $+$ in this context to refer to $\boldsymbol{\theta} + \Delta\boldsymbol{\theta} = \{\boldsymbol{\theta}_i : i \in [n] \setminus J\} \cup \{\boldsymbol{\theta}_j + \Delta\boldsymbol{\theta}_j : j \in J\}$; and where $\mathfrak{L}_r$ denotes the set of all LoRAs with rank $r$ *viz.* $\mathfrak{L}_r = \{\boldsymbol{U}_j\boldsymbol{V}_j : (\boldsymbol{U}_j, \boldsymbol{V}_j) \in \mathbb{R}^{u_j \times r} \times \mathbb{R}^{r \times v_j}\}_{j\in J}$. Thus, rather than learning $\prod_{j\in J} u_j v_j$ parameters, we only need to learn $\prod_{j\in J} r(u_j + v_j)$, which, for a sufficiently small $r$, results in training *significantly* fewer parameters.

In practice, we estimate the expectation in Equation (3) with Monte Carlo sampling and solve the outer minimization loop by stochastic gradient descent. The inner optimization loop is already solved by the adversary, and as such, we do not need to solve the inner maximization loop while training the LoRA parameters. The adversaries were trained to find the maxima for each empirical sample in the training set. *N.B.*, an application of Danskin's theorem ensures that gradients evaluated in maximizers of the inner loop provide a valid descent direction for the saddle point problem in the minimax game (Madry et al., 2018).

## 3.2 LoRA-based mitigation for multiple adversaries

Now, suppose that we not only have an adversary but rather multiple ones. In practice, such adversaries may target different vulnerabilities. We model this by introducing a set of $L$ multiple objective functions $\{\mathcal{L}^\ell\}_{\ell=1}^L$ with $\mathcal{L}^\ell \in \mathcal{C}^\alpha(\mathbb{R}^d \times \mathbb{R}^p \times \mathbb{N})$. Then we can update Equation (3) into this new framework as a set of $L$ *independent* minimax games:

$$\min_{\Delta\boldsymbol{\theta}^\ell\in\mathfrak{L}_r} \mathbb{E}_{(\boldsymbol{X},Y)\sim\mathbb{P}_{\text{data}}} \left[ \mathcal{L}^\ell\left( \boldsymbol{\theta} + \Delta\boldsymbol{\theta}^\ell, \boldsymbol{X} + \arg\max_{\delta\in\mathcal{S}} \mathcal{L}^\ell(\boldsymbol{\theta}, \boldsymbol{X} + \delta, Y), Y \right) \right] \qquad \ell \in [L], \tag{4}$$

where we use the superscript notation to denote objects belonging to a particular threat model for objective $\mathcal{L}^\ell$. Within this context, the fully hardened model parameters are given by

$$\boldsymbol{\theta}^* = \boldsymbol{\theta} + \sum_{\ell=1}^L \pi_\ell \Delta\boldsymbol{\theta}^\ell, \tag{5}$$

where $\sum_\ell \pi_\ell = 1$, $\pi_\ell > 0$ are mixing weights. Clearly, this naïve approach is likely to perform poorly, as LoRA's $\Delta\boldsymbol{\theta}^\ell$ for each vulnerability $\ell$ are unlikely to be orthogonal to one another and may impede performance; this is a well-observed phenomenon in multi-concept generation with LoRAs (Po et al., 2024).

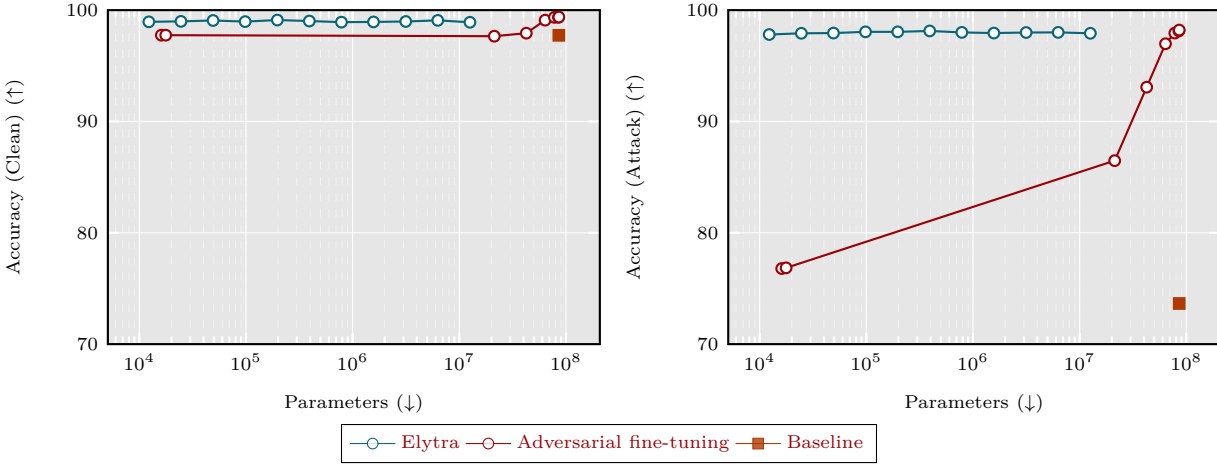

Figure 3: An illustration of accuracy versus parameter count of Elytra and adversarial hardening methods compared to the baseline model. The left graph is the accuracy of the model on non-adversarial data after Elytra or fine-tuning. The right graph is the accuracy of the model on adversarial data after Elytra (blue circle) or fine-tuning (red circle). The y-axis scale is shared to illustrate the model's vulnerability to adversarial inputs. In each graph, the higher and to the left, the better.

### 3.2.1 Continual learning via the composition of LoRAs

Rather than training these concepts in parallel as shown in Equation (4) or imposing an orthogonality constraint between the different adapters *à la* Po et al. (2024); Liang & Li (2024), an alternative is simply to train these adapters sequentially. *I.e.*, we can simply reformulate Equation (4) to be

$$\min_{\Delta\boldsymbol{\theta}^\ell\in\mathfrak{L}_r} \mathbb{E}_{(\boldsymbol{X},Y)\sim\mathbb{P}_{\text{data}}}\left[\mathcal{L}^\ell\left(\boldsymbol{\theta}+\sum_{i=1}^{\ell-1}\Delta\boldsymbol{\theta}^i+\Delta\boldsymbol{\theta}^\ell,\boldsymbol{X}+\arg\max_{\delta\in\mathcal{S}}\mathcal{L}^\ell(\boldsymbol{\theta},\boldsymbol{X}+\delta,Y),Y\right)\right] \qquad \ell\in[L], \qquad (6)$$

and write

$$\boldsymbol{\theta}^* = \boldsymbol{\theta} + \sum_{\ell=1}^{L}\Delta\boldsymbol{\theta}^\ell. \qquad (7)$$

As our primary concern is with the rollout of security adapters in a sequential manner to existing networks, this restriction of the need to train these adapters sequentially is not a significant drawback for our interests. Thus, we can *overcome* the issue of catastrophic forgetting without using expensive complex schemes to orthogonalize the different adapters (Po et al., 2024; Liang & Li, 2024), which *often* requires future knowledge of the components we wish to compose. Moreover, our experimental results in Section 5.3.2 corroborate the utility of this approach in our scenario.

## 4 Experimental setup

To evaluate the validity of the Elytra framework, we conducted experiments using well-known and widely adopted ViT models, Google ViT B/16 (86M parameters) (Dosovitskiy et al., 2021) and Microsoft Swin B/16 (88M parameters) (Liu et al., 2021b), against several adversarial attacks. Both ViT models were pre-trained on the ImageNet-22K dataset (Deng et al., 2009). We further fine-tune these models for the task of classifying road signs on the datasets in Section 4.2. More details on this are provided in Appendix A.2.

When training LoRA adapters, we freeze the entire fine-tuned model before selecting layers and blocks to train. For each of the ViT models, we apply the LoRA adapter to the multi-head attention blocks, multi-layer perceptrons, and classification head.

## 4.1 Adversarial attacks

Following Pavlitska et al. (2023); Zhao et al. (2025), we focus on adversarial attacks in the digital space. As mentioned earlier in Section 3.1.2, we can model adversarial attacks abstractly using the minimax game in Equation (2). In this work, we consider four representative attacks: *Fast Gradient Sign Method* (FGSM) (Goodfellow et al., 2014), *Projection Gradient Descent* (PGD) (Madry et al., 2018; Chen & Hsieh, 2022), *Patch Attack* (Brown et al. (2017)) and *AutoAttack* (Croce & Hein (2020b)).

**FGSM.** Given a single empirical sample $(\boldsymbol{x}, y)$ drawn from our data distribution, the FGSM method (Goodfellow et al., 2014) is a one-step iterative process and is defined as the following map

$$\boldsymbol{x} \mapsto \boldsymbol{x} + \varepsilon \text{sign}(\nabla_{\boldsymbol{x}} \mathcal{L}(\theta, \boldsymbol{x}, y)), \tag{8}$$

where $\varepsilon > 0$ controls the strength of the perturbations (this is typically kept small) and $\text{sign}(\cdot)$ denotes the sign function. *N.B.*, the sign function eliminates the need to explicitly compute gradient norms. Moreover, using the sign function constrains the perturbations to lie within the $\ell^\infty$-ball centered at $\boldsymbol{x}$, which improves the stability of the attack. The adoption of the $\ell^\infty$ (max norm) constraint is not fundamental, since all $\ell^p$ norms are mathematically equivalent in finite-dimensional spaces (Conrad, 2020).

**PGD.** Instead of a single-step update, FGSM can be extended into an iterative algorithm. However, the resulting trajectory $\{\boldsymbol{x}_i\}_{i=1}^n$ may wander outside the feasibility set $\mathcal{S}$. To prevent this, we can introduce a projection step to ensure that iterate $\boldsymbol{x}_i$ stays *within* $\mathcal{S}$. Applying this projection to the iterative FGSM update results in the PGD scheme:

$$\boldsymbol{x}_{i+1} = \Pi_{\mathcal{S}} \left( \boldsymbol{x}_i + \varepsilon \text{sign}(\nabla_{\boldsymbol{x}} \mathcal{L}(\theta, \boldsymbol{x}_i, y)) \right), \tag{9}$$

where $\Pi_{\mathcal{S}}$ denotes the Euclidean projection onto $\mathcal{S}$ and $\boldsymbol{x}_0 = \boldsymbol{x}$. *N.B.*, when the feasibility set $\mathcal{S}$ is chosen as the $\ell^\infty$-ball of radius $\varepsilon$ centered at $\boldsymbol{x}$, the projection reduces to a simple clipping operation; in particular, we clip $\boldsymbol{x}_{i+1}$ into the $\ell^\infty$-ball of radius $\varepsilon$ centered at $\boldsymbol{x}$.

**Patch attack.** Brown et al. (2017) proposed adversarial patch which represents a physical adversarial threat designed to model real-world perturbations such as obstructions or stickers. Unlike traditional adversarial noise, which modifies the entire image and is typically not realizable in the physical world, patch attacks confine the perturbation to a localized region – a "patch" – which can be printed and physically placed within a scene (*e.g.* in a traffic sign). The objective is to cause a misclassification whenever the patch appears within the camera's field of view (FoV), regardless of its position, scale, or orientation.

To achieve this, the attack optimizes local patches for robustness to transformations. During the generation phase, the patch is trained by applying random rotations, scaling, and non-affine distortions to the patch before it is overlaid onto the target image. This process designs the patch to be effective under the natural variability of real-world placements, such as a sticker or obstruction at an angle or viewed from different distances. By optimizing for the robustness of the transformation, the resulting patch becomes a realistic and potent threat.

In this implementation, two patch geometries are utilized: circle and square. Both patches share the same underlying optimization procedure and are parameterized as a tensor of shape and optimized using a cross-entropy loss to target general misclassifications to degrade model performance. Two different patch shapes were selected to simulate different types of obstruction on traffic signs. First, a patch is trained and generated on a subset of data for each split (Validation, Train, or Test), using Adam optimizer to minimize the loss under randomly transformed placements. Then the patch is scaled, rotated, and applied at a random location on each image using the patch specific to the dataset split for which the patch was trained.

**AutoAttack.** Croce & Hein (2020b) proposed AutoAttack, which is an ensemble that combines multiple complementary attacks to give a reliable means to evaluate robustness. It sequentially applies four attacks: APGD-CE (Auto Projected Gradient Descent - Cross Entropy), APGD-DLR (Auto Projected Gradient Descent - Difference of Logits Ratio), FAB (Fast Adaptive Boundary), and Square Attack.

a) APGD is a parameter-free version of the PGD attack that dynamically adjusts the step size. For APGD, Equation (9) is redefined as

$$z_{i+1} = \Pi_{\mathcal{S}}\left(x_i + \varepsilon_i \text{sign}(\nabla_x \mathcal{L}(\theta, x_i, y))\right), \tag{10}$$

$$x_{i+1} = \Pi_{\mathcal{S}}\left(x_i + \alpha(z_{i+1} - x_i) + (1 - \alpha)(x_i - x_{i-1})\right) \tag{11}$$

where $z_i$ acts as a momentum term and $\alpha$ controls the interpolation between updates. The step size $\varepsilon_{i+1}$ is adaptively halved whenever the loss does not increase in the $i+1$ iteration, and the algorithm reverts to a previous checkpoint to maintain progress.

APGD-DLR further replaces the cross-entropy loss with the Difference of Logits Ratio (DLR), in order to maximize the margin between the correct class's logit and all other class logits.

c) FAB attack (Croce & Hein, 2020a) is designed to find the minimal $\ell_p$-norm perturbation of the linearized decision boundary between the target class and the correct class. It employs a geometric approach by approximating the decision boundary linearly and computing the smallest adversarial perturbation needed to cross it.

d) Square attack is an adversarial attack method based on randomized search (Croce et al., 2019). It perturbs localized square-shaped regions of the input, along with a specific initialization scheme using queries on the model's final output.

The combination of gradient-based and non-gradient-based attacks makes it particularly effective against models that may be robust against either attack.

## 4.2   Datasets

For this study, we chose the Mapillary traffic sign dataset (Neuhold et al., 2017), a large collection of street-level imagery with annotations including bounding boxes, semantic segmentation, and attribute tags covering more than 100 categories.

We selected 21 classes of interest, defined in Table 1, that are both prevalent in real-world driving scenarios and well represented in the dataset. These classes encompass regulatory signs (*e.g.* speed limits, stop, no entry), warning signs (*e.g.* curve), and information signs (*e.g.* keep left, parking). The selection criteria prioritized classes with both diversity in sign quality and sufficient samples to support robust training and evaluation of vision transformers in the presence of adversarial attacks.

The images in the original dataset varied in resolution and contained complex street scenes. To isolate desired traffic signs, we leveraged the provided bounding box annotations to crop each sign tightly. Then, all cropped images were resized and cropped if needed to $224 \times 224$ pixels using bilinear interpolation, matching the input size expected of the selected vision transformers mentioned at the beginning of Section 4. This standardization ensures consistency when the input is passed to vision transformers and when adversarial samples are generated.

After filtering using the defined selection criteria, the final dataset is made up of 56,521 images, distributed over 21 classes, as detailed in Table 1. Prior to training the vision transformers on the data, the data was partitioned into training (70%), validation (15%), and test (15%) splits, stratified by class to maintain class balance and sufficient samples for validation and testing. Additionally, to ensure that data leakage does not occur, stratified splits were stored and accessed in csv files that contain the class, image source, and dataset.

## 5   Results

### 5.1   Comparison with full fine-tuning

We performed an ablation study to validate both the choice of LoRA rank and the validity of LoRA over adversarial hardening. For this study, a single attack was used against Google ViT B/16 after fine-tuning the model on Mapillary data. AutoAttack was chosen for its adversarial strength to illustrate the impact

Table 1: Dataset Class Breakdown

| Class | Samples | Class | Samples |
|---|---|---|---|
| Ahead Only | 1,438 | No Right Turn | 1,004 |
| Curve | 2,596 | No Stopping | 3,079 |
| Goods Vehicles | 587 | No U-Turn | 1,297 |
| Keep Left | 1,345 | Parking | 2,690 |
| Keep Right | 2,691 | Priority Road | 1,657 |
| No Entry | 2,089 | Roundabout | 1,827 |
| No Left Turn | 1,231 | Speed Limit | 14,910 |
| No Overtaking | 1,613 | Stop | 2,552 |
| No Parking | 3,518 | Turn Left | 2,016 |

LoRA rank has on the classification accuracy of adversarial and original test data. Different degrees of adversarial hardening were performed to present a baseline using standard practices for adversarial robustness and plotted in Table 10 and Figure 3.

Table 2: Comparison of security strategies across models (ViT and Swin) on adversarial and clean samples. Values shown as ViT/Swin. Higher is better.

| Model | Clean | FGSM | PGD | Square | Circle | AutoAttack |
|---|---|---|---|---|---|---|
| Baseline | 97.74/97.68 | 89.76/92.88 | 75.14/89.01 | 89.14/76.32 | 94.33/84.48 | 73.65/83.37 |
| FGSM LoRA | **99.25**/99.34 | **98.27**/**99.11** | 92.28/98.77 | 91.83/81.28 | 96.66/87.65 | 90.70/97.54 |
| PGD LoRA | 98.95/99.25 | 96.35/98.66 | 98.38/99.05 | 93.93/86.65 | 97.48/91.70 | 96.95/98.34 |
| Square Patch LoRA | 98.98/**99.49** | 92.66/97.37 | 75.84/95.48 | **98.45**/**98.83** | 98.72/**99.12** | 75.38/92.05 |
| Circle Patch LoRA | 99.07/99.37 | 93.41/97.17 | 78.21/95.30 | 98.30/98.65 | **98.77**/98.97 | 77.50/91.76 |
| AutoAttack LoRA | 98.99/98.96 | 97.59/98.88 | **98.51**/**99.24** | 93.57/90.41 | 97.28/94.10 | **97.92**/**98.98** |

The single Elytra patch models all performed better than the original fine-tuned models on both adversarial data and "clean", i.e., unedited data. Importantly, the rank of the Elytrahas little impact on the classification accuracy of adversarial data between different Elytras and has a negligible impact on the accuracy of clean data. The best-performing rank in this ablation is $r = 16$, so we will use this rank for all Elytras going forward. Unsurprisingly, models fine-tuned with traditional adversarial hardening methods show that the fewer parameters that are frozen, the better the model performs on adversarial data. This is not without its drawbacks, however, as the more unfrozen parameters there are, the longer the model takes to train and fine-tune. The model hardened with only one layer frozen performs as well as the Elytra evaluated. Similarly, the model hardened with three frozen layers performs similarly to the LoRA selected on clean data. This result is in alignment with previous work (Pan et al., 2024) that found a consistent skew in the contributions of the weight-norm per-layer during fine-tuning, with the input and output layers dominating the parameter updates. The difference between a LoRA with $r = 16$ and a model with three frozen layers is 63.6 million parameters; thus, we achieve a reduction of 99.7% trainable parameters while maintaining adversarial robustness for a fraction of training time and resources.

## 5.2 On the generalization of a single Elytra patch

Now that we have established the effectiveness of LoRA over the other benchmarks with adversarial samples under AutoAttack, we seek to explore the effectiveness of LoRA for other adversarial samples. To further probe LoRA's effectiveness, the question we want to ask is whether training an Elytra on adversarial data from any specific attack makes the updated model robust to that attack, *viz.*, *can LoRA act as a targeted defense mechanism*? To test this, we trained separate Elytras for each of the adversarial attacks and then measured the performance of the model before and after implementing an Elytra. The result of this experiment is shown in Table 2.

For each model and each adversarial attack, we measured model performance before and after updating with an Elytra trained on adversarial samples. We also record model performance on clean samples before

and after the Elytra is trained on adversarial samples. A small change in the model performance on clean samples before and after, followed by a large increase in model performance on each adversarial testing sample following adversarial training, indicates model robustness towards that attack. While a single Elytra is able to defend against a singular targeted attack, models require generalized robustness against a range of diverse attacks in order to operate properly in the presence of adversarial inputs.

### 5.3 On the composability of multiple Elytra patches

As observed in Section 5.2 we find that a single Eltyra adequately patches the targeted vulnerability; however, we have also noticed that it is not our *panacea*. This is quite reasonable, as security patches are often targeted. The next question then is how to apply multiple patches to our network? We begin with the naïve approach of training several Elytras in parallel *à la* Equation (4); and then secondly we exploit the structure of rolling out security patches to train Elytras in sequence *à la* Equation (6).

#### 5.3.1 Parallel training

We begin by exploring the naïve approach outlined in Equation (4), *i.e.*, simply training multiple Elytras in parallel in a blind manner, *viz.*, not considering the other attacks. We begin by composing two patches together in Table 3 and increasing the number of patches in subsequent tables, which will be discussed later.

Table 3: Performance comparison of dual Elytra combinations across architectures (ViT/Swin). Higher is better.

| Model | Clean | FGSM | PGD | Square | Circle | AutoAttack |
|---|---|---|---|---|---|---|
| Baseline | 97.74/97.68 | 89.76/92.88 | 75.14/89.01 | 89.14/76.32 | 94.33/84.48 | 73.65/83.37 |
| Circle+Square | 99.18/99.09 | 95.26/97.55 | 74.18/94.19 | 98.10/98.47 | 98.77/98.32 | 74.69/89.96 |
| Circle+PGD | 99.29/99.33 | 97.79/98.50 | **97.98**/98.00 | 98.18/97.98 | 98.81/98.43 | 96.23/96.97 |
| Circle+FGSM | **99.33**/99.08 | 98.30/97.53 | 89.19/94.90 | 98.06/97.33 | 98.69/97.75 | 88.13/92.48 |
| Circle+AutoAttack | 99.24/99.33 | 98.19/98.91 | 97.91/98.51 | 98.10/98.36 | 98.72/98.64 | **97.10**/98.15 |
| Square+PGD | 99.29/**99.46** | 97.73/98.86 | 97.84/98.77 | 98.34/98.57 | **98.98**/98.73 | 96.30/97.76 |
| Square+FGSM | 99.23/99.36 | 98.32/98.56 | 87.57/96.88 | 98.18/98.26 | 98.79/98.34 | 86.29/95.04 |
| Square+AutoAttack | 99.28/99.44 | 98.18/**99.10** | 97.71/**98.83** | **98.45/98.68** | 98.92/**98.83** | 96.92/**98.44** |
| PGD+FGSM | 99.10/99.25 | 98.46/98.70 | 96.53/98.02 | 93.89/87.88 | 97.26/92.44 | 95.10/97.24 |
| PGD+AutoAttack | 98.56/99.16 | 97.97/98.96 | 96.11/98.42 | 92.19/94.24 | 95.41/96.14 | 94.81/98.19 |
| FGSM+AutoAttack | 99.01/99.20 | **98.58**/98.72 | 96.92/98.30 | 93.55/93.12 | 97.15/95.81 | 96.44/97.81 |

In Table 3 we composed two patches simply and notice that the performance improves. In particular, the circle, square, and PGD composed Elytra achieve the best generalization, whereas the PGD, FGSM, and AutoAttack compositions show a new trend not previously observed: an overall degradation in performance. That model performs worse than its single-Elytra counterparts at 9.44% to 10.11% below the baseline, indicating parameter interference in the adapter matrix space.

Table 4: Performance comparison of triple LoRA combinations across architectures (ViT/Swin). Higher is better.

| Model | Clean | FGSM | PGD | Square | Circle | AutoAttack |
|---|---|---|---|---|---|---|
| Baseline | 97.74/**97.68** | 89.76/92.88 | 75.14/89.01 | 89.14/76.32 | 94.33/84.48 | 73.65/83.37 |
| Circle+Square+PGD | **99.04**/94.23 | 97.27/90.33 | **95.28**/82.29 | **97.33**/90.61 | **98.19**/91.89 | **92.64**/78.53 |
| Circle+Square+FGSM | 98.72/93.26 | 96.81/84.25 | 77.66/73.29 | 95.77/90.31 | 97.23/90.89 | 78.46/68.59 |
| Circle+Square+AA | 98.86/94.97 | 97.07/91.93 | 93.51/84.49 | 96.78/92.05 | 97.74/93.18 | 92.18/82.60 |
| Circle+PGD+FGSM | 98.76/93.92 | 97.61/89.56 | 92.88/86.40 | 95.38/85.69 | 96.90/88.44 | 91.51/83.96 |
| Circle+PGD+AA | 96.76/96.39 | 95.74/95.22 | 92.05/94.06 | 91.07/90.14 | 93.18/92.37 | 90.49/93.23 |
| Circle+FGSM+AA | 98.05/95.75 | 96.92/93.08 | 89.64/91.78 | 94.09/88.73 | 95.96/91.13 | 88.63/90.75 |
| Square+PGD+FGSM | 98.77/96.53 | **97.85**/93.53 | 93.35/90.62 | 96.18/90.80 | 97.17/92.31 | 91.93/89.24 |
| Square+PGD+AA | 97.37/97.65 | 96.42/**96.96** | 93.26/**95.52** | 92.86/**93.48** | 94.55/**95.01** | 91.91/**94.73** |
| Square+FGSM+AA | 98.13/97.28 | 97.14/95.69 | 91.52/94.10 | 95.27/92.60 | 96.45/94.09 | 90.67/93.06 |
| PGD+FGSM+AA | 92.10/95.86 | 90.94/95.06 | 81.12/89.98 | 79.70/86.87 | 84.22/90.55 | 81.27/88.05 |

At four and five Elytras, performance is universally degraded. The best composition of four attacks sees a 1.84% drop in classification accuracy, while the worst performing composition plummets 18.65% below the

baseline on clean data and 24.09% on adversarial samples. Composing all five Elytras represents a complete catastrophic failure, performing 38.44% worse than the baseline on clean data and performing between 19.68% and 39.92% worse on adversarial inputs. This trend of performance degradation is visualized in Figure 4. This leads to the conclusion that parallel training is neither effective nor capable of composition without extensive design prior to composition to maintain parameter space orthogonalization.

Table 5: Performance comparison of quadruple and quintuple Elytra combinations across architectures (ViT/Swin). Higher is better.

| Model | Clean | FGSM | PGD | Square | Circle | AutoAttack |
|---|---|---|---|---|---|---|
| Cir+Sq+PGD+FGSM | 95.90/73.54 | **93.49**/66.86 | 79.69/61.77 | **89.93**/68.21 | 92.44/69.30 | 79.33/59.21 |
| Cir+Sq+PGD+AA | 90.47/73.15 | 88.68/66.73 | **83.18**/61.79 | 83.55/65.94 | 85.58/68.52 | **81.85**/60.37 |
| Cir+Sq+FGSM+AA | 92.05/68.20 | 87.50/59.65 | 68.42/55.25 | 85.13/62.74 | 87.17/64.31 | 70.42/53.80 |
| Cir+PGD+FGSM+AA | 79.09/81.03 | 76.32/83.91 | 66.17/77.84 | 68.11/69.08 | 71.24/71.72 | 67.05/75.07 |
| Sq+PGD+FGSM+AA | 83.37/86.01 | 81.87/87.01 | 72.37/80.51 | 73.58/75.00 | 75.98/78.11 | 72.57/77.50 |
| All 5 Combined | 59.30/45.70 | 59.15/51.77 | 53.09/48.73 | 52.31/38.37 | 54.41/40.06 | 53.97/46.19 |
| Baseline | **97.74**/**97.68** | 89.76/**92.88** | 75.14/**89.01** | 89.14/**76.32** | **94.33**/**84.48** | 73.65/**83.37** |

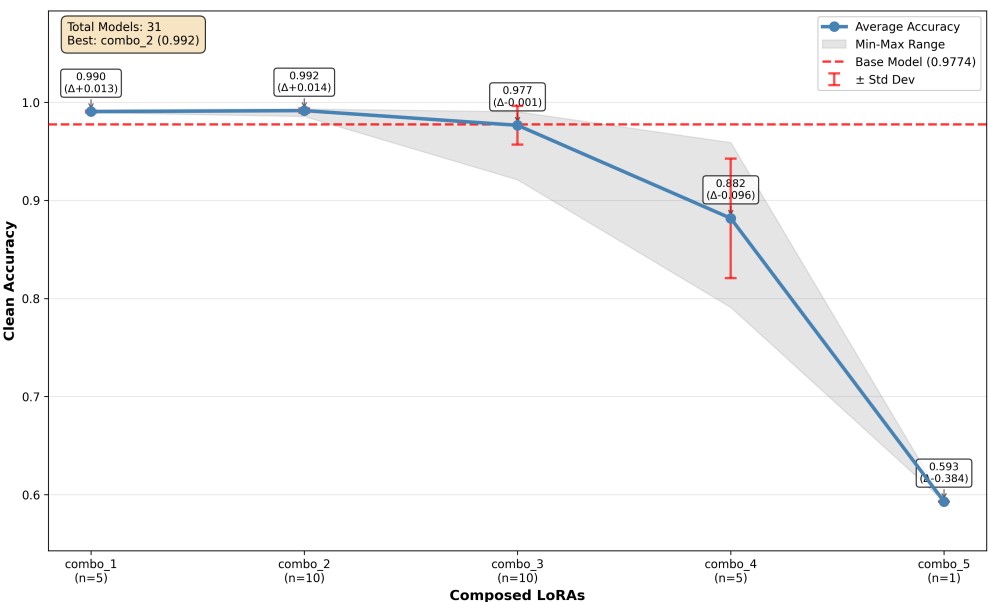

Figure 4: Performance of Elytras as the quantity of Elytras composed increases. n represents the number of models generated per number of Elytras composed

### 5.3.2 Sequential training

As established in the previous section, simply averaging the parameters of multiple Elytras leads to catastrophic interference and model failure, particularly as more Elytras are combined; see Figure 4, Figure 9, and Figure 10. This behavior indicates that the parameter spaces of independently trained Elytras are not directly alignable through simple merging. To prevent catastrophic interference, we train Elytras in *sequence*, where each new Elytra is trained on a model that has already been updated by the previous Elytras in the sequence and is frozen. This method allows the model to find a parameter configuration that accommodates multiple adversarial threats.

The results, shown in Table 6, demonstrate the remarkable strength of the sequential approach. Both the standard and the inverse sequences achieve consistently high accuracy across all baselines and attacks on

Table 6: Performance comparison of Elytra combinations trained sequentially across architectures (ViT/Swin). Values shown as ViT/Swin. Higher is better. Sequence One: Patch Circle → PGD → Patch Square → FGSM → AutoAttack. Sequence Two (Inverse Order of Sequence One): AutoAttack → FGSM → Patch Square → PGD → Patch Circle.

| Model | Clean | FGSM | PGD | Square | Circle | AutoAttack |
|---|---|---|---|---|---|---|
| Sequence One | 99.42/99.56 | **98.71**/99.31 | 98.82/**99.31** | **98.70**/98.79 | 99.18/99.18 | **98.37/99.12** |
| Sequence Two | **99.46/99.57** | 98.68/**99.37** | **98.94**/99.24 | 98.68/**99.21** | **99.22/99.38** | 98.24/98.98 |
| Baseline | 97.74/97.68 | 89.76/92.88 | 77.61/89.01 | 89.14/76.32 | 94.33/84.48 | 73.65/83.37 |

both sets of architectures. Crucially, the near-identical performance of Sequence One and Sequence Two indicates that the effectiveness is independent of the order where the sequential training process itself is key to successful composition.

## 6 Conclusion

In this work, we propose ELYTRA as a solution to rolling out lightweight security patches for large pre-existing vision systems and, in particular, as security patches for components of autonomous driving systems. As vision models continue to grow larger – consider Google's recent 22B parameter ViT (Dehghani et al., 2023) – this proposed framework will only grow increasingly more relevant. We studied the use case of ViTs trained to classify traffic signs under the attack of adversarial examples. Our results on hardening ViTs against adversarial attacks show that the proposed framework has real promise for solving the reliability and safety concerns of autonomous vision systems. We successfully demonstrated that the ELYTRA framework can roll-out multiple security patches against multiple exploits whilst maintaining robustness. We empirically show that this sequential nature instinctively lends itself to avoiding the problem of catastrophic forgetting common to other implementations of LoRA compositions in other systems; thereby enabling us to avoid expensive orthogonalization procedures or procedures which require *future* knowledge.

**Limitations and future work**

While our results are very promising, there are several facets we have yet to consider, namely, improving the composability of multiple patches, hardening against non-digital attacks to adversarial examples, and considering multiple types of vision systems. We believe that these would make for excellent future work, but the lack thereof does not detract from the contributions of this paper.

**Broader impact statement**

Whilst we believe that the downstream applications of our work are mostly positive, *i.e.*, a security mechanism for hardening large vision systems against adversarial attacks, we acknowledge that any security technology carries potential dual-use implications. By enabling targeted security patches with far fewer parameters, this enables organizations with limited computational resources to protect safety-critical systems such as autonomous vehicles. This capability for rapid, lightweight patching could significantly reduce the exposure window for vulnerabilities as they are discovered. Conversely, the same methodology could be used to efficiently inject backdoors rather than patch them, though this will not enable the deployment of new attacks, as this requires the same level of access as current backdoor attacks. Furthermore, the data examined in this paper do not cover real-world representations of the digital attacks, so the domain transfer cannot be verified without further work. We believe that the benefits of enabling accessible, efficient security hardening – particularly for autonomous systems where misclassification of traffic signs could have severe consequences – substantially outweigh the aforementioned risks.

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

# A    Experimental details

**Hardware.**    All computations and evaluations were performed on a system composed of a Ryzen 7 5800x3D CPU and a NVIDIA GeForce RTX 4090 with CUDA version 12.8 and CUDNN 9.10.2. The proposed framework was implemented in PyTorch.

## A.1    Adversarial Generation

For both gradient attacks, FGSM and PGD, torchattacks is used to implement the attacks using the FGSM and PGD classes. For both Google ViT and Swin, an $\epsilon$ value of $\frac{8}{255}$ is selected as the starting point. PGD increments this by $\frac{2}{255}$ for 10 iterations. For AutoAttack, an $\epsilon$ value of $\frac{8}{255}$ is selected as the starting point.

The patch attacks, Circle and Square, utilize the AdversarialPatchPyTorch class from ART (Adversarial-Robustness-Toolbox) [3]. Attacks optimize patch patterns over 500 iterations using Adam optimization with a learning rate of 5. Both circle and square patches are trained with random scaling $(0.05 - 1.0)$ and rotation $(\pm22.5°)$. During application, patches are randomly positioned and scaled $(0.1 - 0.5$ of image dimensions) to simulate real-world patch and debris placement over the sign area.

## A.2    Fine-tuning the ViTs

The following designs are implemented during training to improve performance. The AdamW optimizer(Kingma & Ba, 2014) with weight decay $(1 \times 10^{-4})$ was used for fine-tuning all models. All fine-tuned models were trained using a step learning rate scheduler starting at $1 \times 10^{-4}$ for initial fine-tuning and $1 \times 10^{-5}$ for adversarial fine-tuning, with step sizes varying by training approach. Data were pre-processed with normalization to rescale input features to a standardized range before training. This ensures that the gradients propagate evenly through the layers, resulting in a more robust and accurate model (Ioffe & Szegedy, 2015; Kornblith et al., 2019). The normalization for Google B/16 ViT was based on the values of Google's Hugging Face model card[4] (Wu et al., 2020). The normalization for Swin-B is derived from ImageNet normalization standard values[5] (Liu et al., 2021b; Paszke, 2019; Deng et al., 2009).

Table 7: Initial Fine-tuning Hyperparameters

| Hyperparameter | Google B/16 | Swin-B |
|---|---|---|
| Image size | 224 | 224 |
| Batch size | 32 | 32 |
| Learning rate | $1 \times 10^{-4}$ | $1 \times 10^{-4}$ |
| Weight decay | $1 \times 10^{-4}$ | $1 \times 10^{-4}$ |
| Epochs | 24 | 24 |
| Learning rate decay | StepLR (step=20, $\gamma = 0.1$) | StepLR (step=20, $\gamma = 0.1$) |
| Optimizer | AdamW | AdamW |
| Normalization ($\mu$) | [0.5, 0.5, 0.5] | [0.485, 0.456, 0.406] |
| Normalization ($\sigma$) | [0.5, 0.5, 0.5] | [0.284, 0.262, 0.284] |

Table 8: Adversarial Fine-tuning Hyperparameters

| Hyperparameter | Google B/16 | Swin-B |
|---|---|---|
| Image size | 224 | 224 |
| Batch size | 64 | 64 |
| Learning rate | $1 \times 10^{-5}$ | $1 \times 10^{-5}$ |
| Fine-tuning LR | $1 \times 10^{-6}$ | $1 \times 10^{-6}$ |
| Weight decay | $1 \times 10^{-4}$ | $1 \times 10^{-4}$ |
| Epochs | 12 | 12 |
| Freeze strategy | 8 variants | 8 variants |
| Adversarial attacks | 5 types | 5 types |
| Learning rate decay | StepLR (step=10, $\gamma = 0.5$) | StepLR (step=10, $\gamma = 0.5$) |

---

[3] https://github.com/Trusted-AI/adversarial-robustness-toolbox
[4] https://huggingface.co/google/vit-base-patch16-224
[5] https://github.com/pytorch/vision/blob/main/torchvision/models/video/swin_transformer.py

### A.3 Training the LoRAs

Following the work done by Bafghi et al. (2024), a LoRA of rank $r = 16$ is selected for adaptation. When adding LoRA to a ViT, all fine-tuned parameters are kept frozen. Doing so allows for updating the attention query, attention value, and classification layer without touching position embeddings, patch embeddings, multilayer perceptron (MLP) blocks, or key projections. The LoRA is trained using adversarial training data with a target to return images causing misclassifications towards their true class, which becomes a security patch after being composed with a fine-tuned model.

Table 9: LoRA Training Hyperparameters

| Hyperparameter | Google B/16 | Swin-B |
|---|---|---|
| Image size | 224 | 224 |
| Batch size | 32 | 32 |
| Learning rate | $1 \times 10^{-4}$ | $1 \times 10^{-4}$ |
| LoRA rank $r$ | 16 | 16 |
| LoRA $\alpha$ | $r \times 2$ | $r \times 2$ |
| LoRA dropout | 0.1 | 0.1 |
| Epochs | 4 | 4 |
| Target modules | query, key, value, output.dense | query, key, value, attention.output.dense |

## B  Additional results

Table 10: Ablation on the impact of fine-tuning strategies on classification accuracy for Google ViT-B/16 on the Mapillary dataset in the presence of AutoAttack adversarial examples. Higher is better.

| | | | Classification accuracy | |
|---|---|---|---|---|
| Model | Layers | Parameters ($\downarrow$) | Clean ($\uparrow$) | Attack ($\uparrow$) |
| LoRA ($r = 1$) | All | 12.3K | 98.95 | 97.81 |
| LoRA ($r = 2$) | All | 24.6K | 98.99 | 97.92 |
| LoRA ($r = 4$) | All | 49.2K | 99.07 | 97.94 |
| LoRA ($r = 8$) | All | 98.3K | 98.97 | 98.05 |
| LoRA ($r = 16$) | All | 197K | **99.11** | 98.05 |
| LoRA ($r = 32$) | All | 393K | 99.02 | **98.13** |
| LoRA ($r = 64$) | All | 786K | 98.92 | 98.00 |
| LoRA ($r = 128$) | All | 1.57M | 98.94 | 97.94 |
| LoRA ($r = 256$) | All | 3.15M | 98.98 | 97.99 |
| LoRA ($r = 512$) | All | 6.29M | 99.08 | 98.00 |
| LoRA ($r = 1024$) | All | 12.6M | 98.91 | 97.92 |
| Adversarial Hardening | Backbone | 16.1K | 97.75 | 76.79 |
| Adversarial Hardening | Encoders | 17.7K | 97.75 | 76.85 |
| Adversarial Hardening | 9 | 21.3M | 97.66 | 86.48 |
| Adversarial Hardening | 6 | 42.5M | 97.93 | 93.08 |
| Adversarial Hardening | 3 | 63.8M | 99.10 | 96.98 |
| Adversarial Hardening | 1 | 78.0M | 99.35 | 97.94 |
| Adversarial Hardening | Embeddings | 85.1M | 99.36 | 98.13 |
| Adversarial Hardening | None | 85.8M | **99.38** | **98.21** |
| Baseline Model | - | 85.8M | 97.74 | 73.65 |

Due to space constraints and clarity, the main paper focuses on the Elytra results. The confusion matrices of Google ViT B/16 and Swin-B, while relevant and interesting, risk distracting from the core findings of the study. Below, the Google and Swin-B confusion matrices for the parameter-averaged adapters and sequentially trained adapters are included.

### B.0.1  ViT parameter averaged adapters

The confusion matrices are calculated on the model before and after applying an adapter to defend against attacks, using both adversarial and non-adversarial, *i.e.*, "clean" data. The average normalized confusion matrix seen in Figure 5 emphasizes the broader impact that adversarial data have on a model's ability to differentiate similar classes. The Google ViT model, in particular, struggles with differentiating between "keep left" and "keep right" as well as "turn left" and "turn right" even before being tested against adversarial

samples. In the presence of adversarial test sets, the models' performance of the aforementioned classes continues to degrade, with additional impacts to performance seen in increasingly dissimilar classes, including but not limited to "no parking", "no stopping", "no right turn", and "goods vehicles".

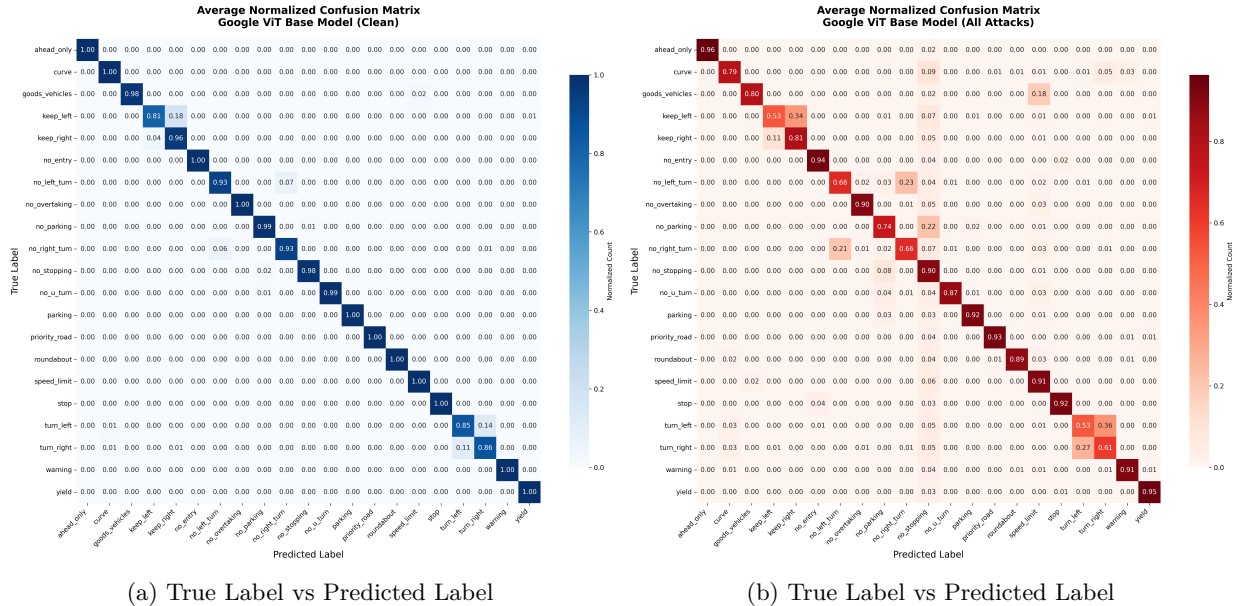

(a) True Label vs Predicted Label          (b) True Label vs Predicted Label

Figure 5: Fine-tuned Google ViT performance: Confusion matrices comparing model performance on (a) clean and (b) adversarial test sets. Adversarial test results are the aggregate results of each attack.

After training adapters on adversarial samples and merging the models previously illustrated in Figure 5, significant performance gains on all classes are observed, further detailed in Figure 6. There are some classes that have decreased performance in the adversarial test sets, albeit still higher performance than models without an adapter trained on adversarial data. The new models, likewise, performed better than the baseline on the clean test set for every class.

The trend observed in Section 5.2 continues when a second adapter trained on different sets of adversarial samples is added and the models are merged. The models with two adapters, Figure 7, show marginal performance gains on most classes in the presence of adversarial samples, with larger performance increases on classes that had classification issues in models with one or no adapters. The models, likewise, performed better than the baseline and single adapter model on the clean test set.

The trend seen in the single and dual adapter models does not continue as a third adapter is composed with parameter averaging. The performance on the non-adversarial data is approximately the same as previous models; however, performance on adversarial data is worse than both the average of the single-adapter models and the dual-adapter models. This point marks the performance limit where simply averaging the adapters together, instead of increasing performance, adding adapters degrades the model's ability to properly classify test sets.

The decline in performance observed in Figure 8 not only continues but also develops into catastrophic failure for models with four and five adapters in most classes. This is most easily observed in Figure 9 and Figure 10, where the accuracy on clean and adversarial data falls below the baseline model performance for the quintuple adapter model and, in most cases, for the quadruple adapter model.

As seen in Figure 11 and Figure 12, the model does not experience catastrophic forgetting and performance drops seen in the parameter-averaged models.

The outcome seen here is in direct contrast to the failure observed in parameter averaging. Compared to catastrophic forgetting and degradation seen in Table 5, sequential training successfully assembles five adapters without collapsing in performance. Furthermore, the sequential model performs on par with or

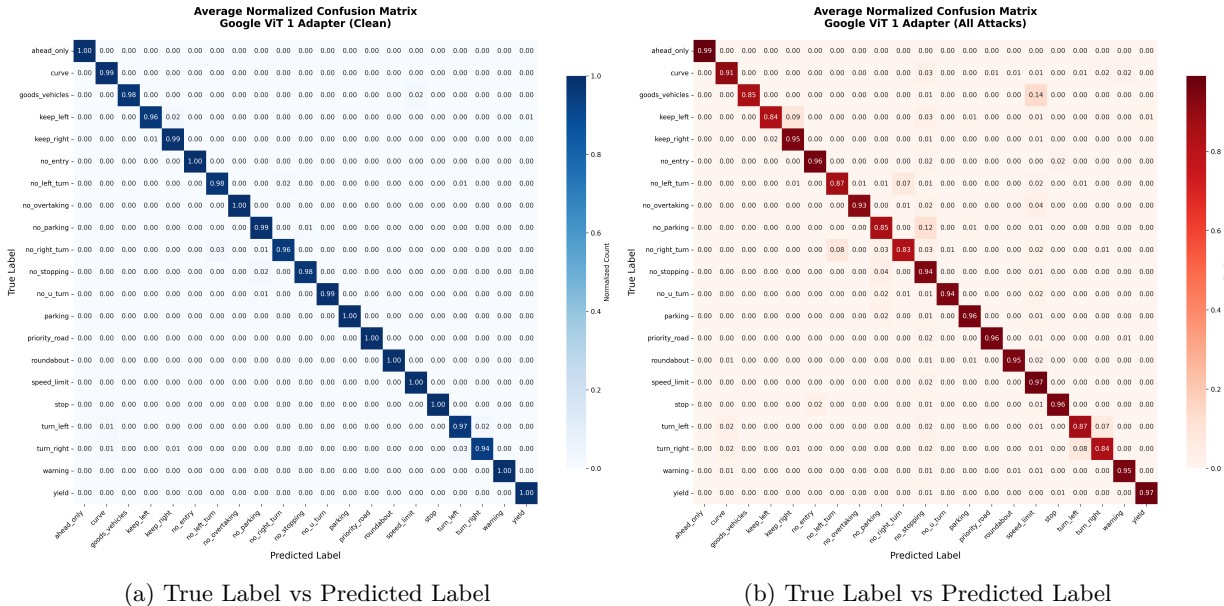

(a) True Label vs Predicted Label

(b) True Label vs Predicted Label

Figure 6: Google ViT with one adapter: Confusion matrices comparing model performance on (a) clean and (b) adversarial test sets. Adversarial and clean test results are the aggregate results of each single adapter merged model. Adversarial test results are the aggregate results of each attack.

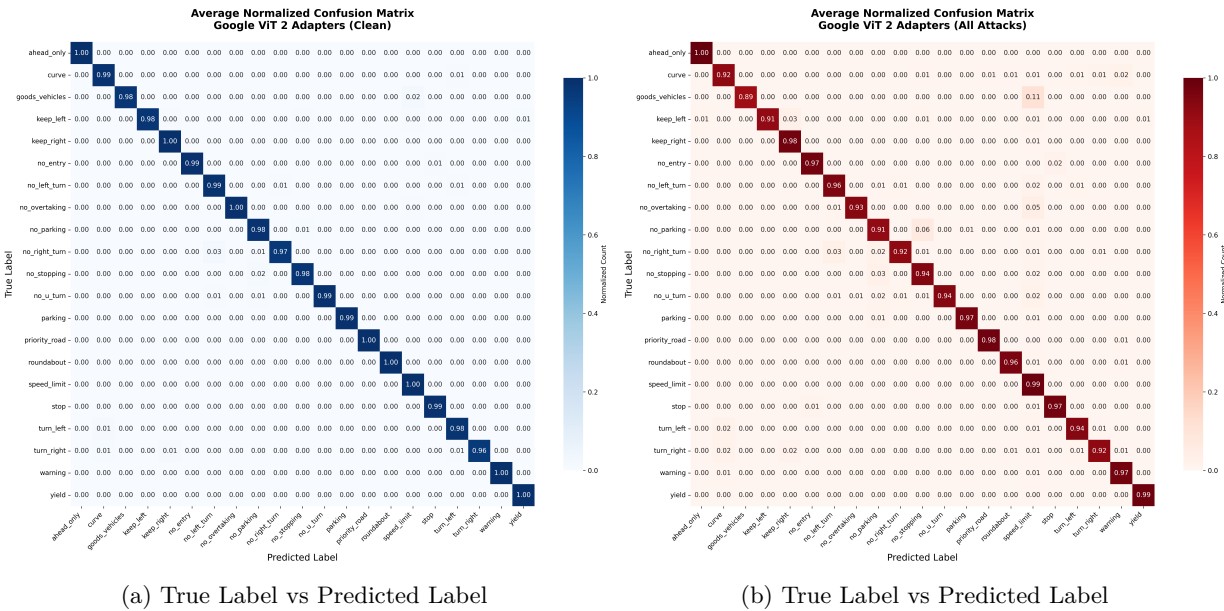

(a) True Label vs Predicted Label

(b) True Label vs Predicted Label

Figure 7: Google ViT with two parameter-averaged adapters performance: Confusion matrices comparing model performance on (a) clean and (b) adversarial test sets. Adversarial and clean test results are the aggregate results of each dual adapter merged model. Adversarial test results are the aggregate results of each attack.

exceeds the best single-adapter from Table 2, but does so for all attack types. This shows that sequential training enables the model to accumulate and retain parameters to be robust against multiple attack types, effectively finding a stable point in the parameter space that generalizes. Figure 11 and Figure 12 show that

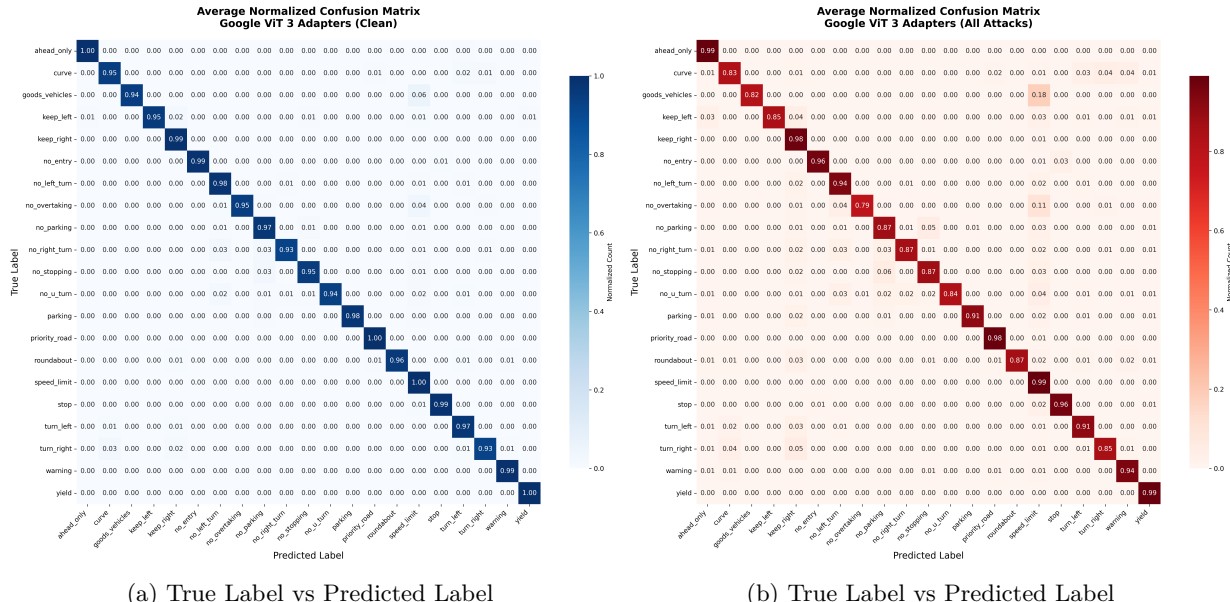

(a) True Label vs Predicted Label  (b) True Label vs Predicted Label

Figure 8: Google ViT with three parameter-averaged adapters performance: Confusion matrices comparing model performance on (a) clean and (b) adversarial test sets. Adversarial and clean test results are the aggregate results of each triple adapter merged model. Adversarial test results are the aggregate results of each attack.

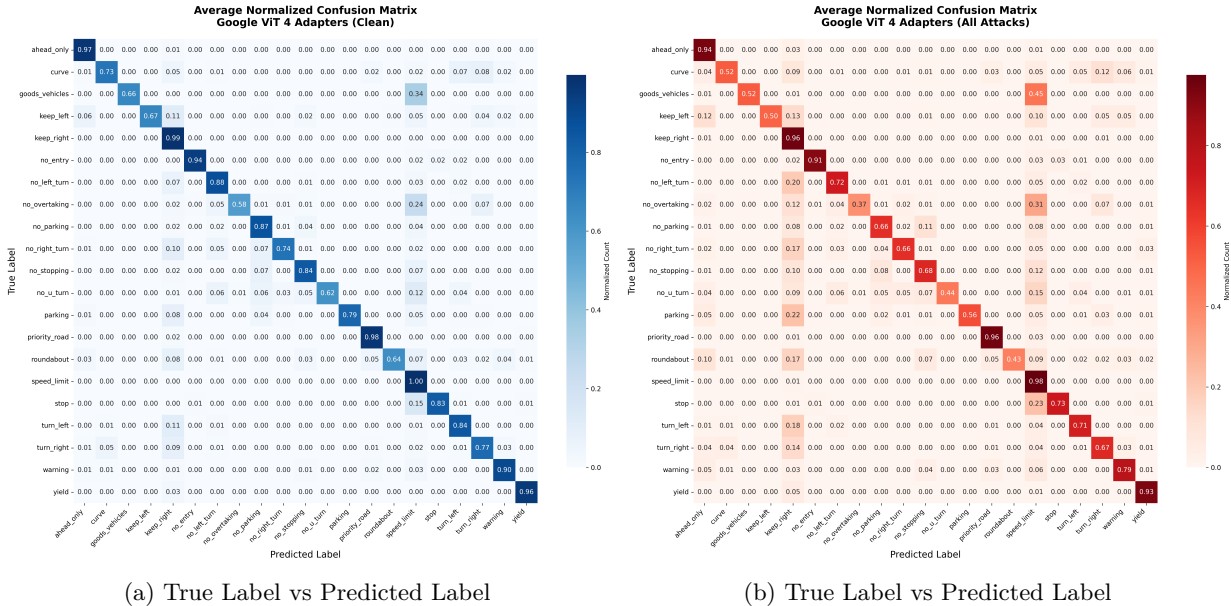

(a) True Label vs Predicted Label  (b) True Label vs Predicted Label

Figure 9: Google ViT with four parameter-averaged adapters performance: Confusion matrices comparing model performance on (a) clean and (b) adversarial test sets. Adversarial and clean test results are the aggregate results of each quadruple adapter merged model. Adversarial test results are the aggregate results of each attack.

the model is able to generalize adversarial and clean data in different training configurations with negligible performance differences.

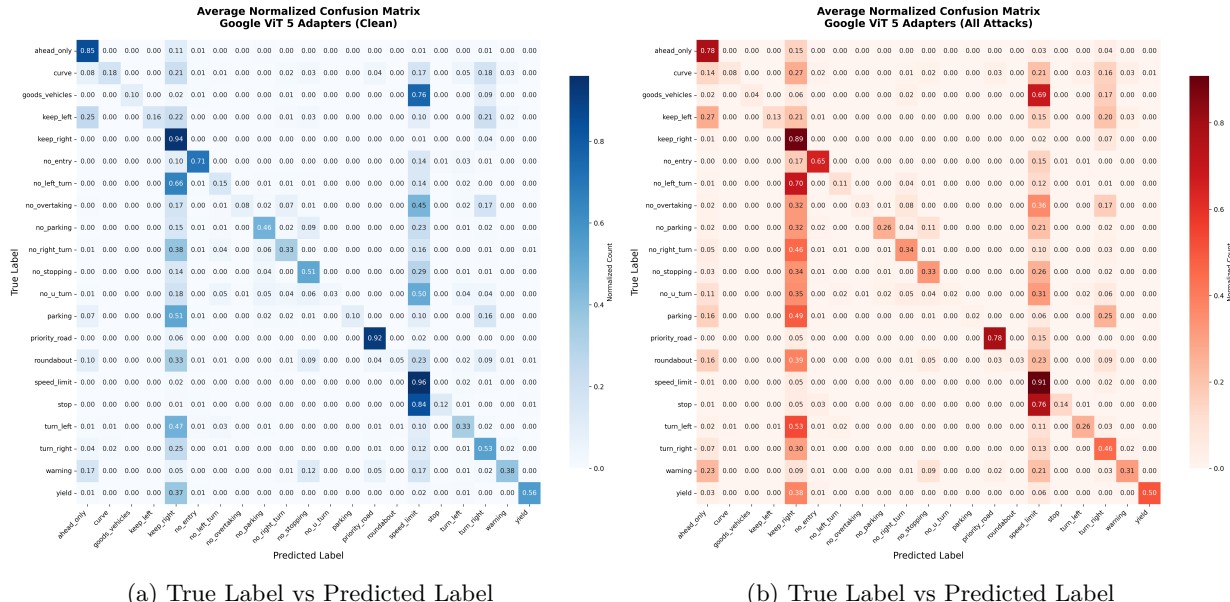

(a) True Label vs Predicted Label    (b) True Label vs Predicted Label

Figure 10: Google ViT with five parameter-averaged adapters performance: Confusion matrices comparing model performance on (a) clean and (b) adversarial test sets. Adversarial and clean test results are the aggregate results of each quintuple adapter merged model. Adversarial test results are the aggregate results of each attack.

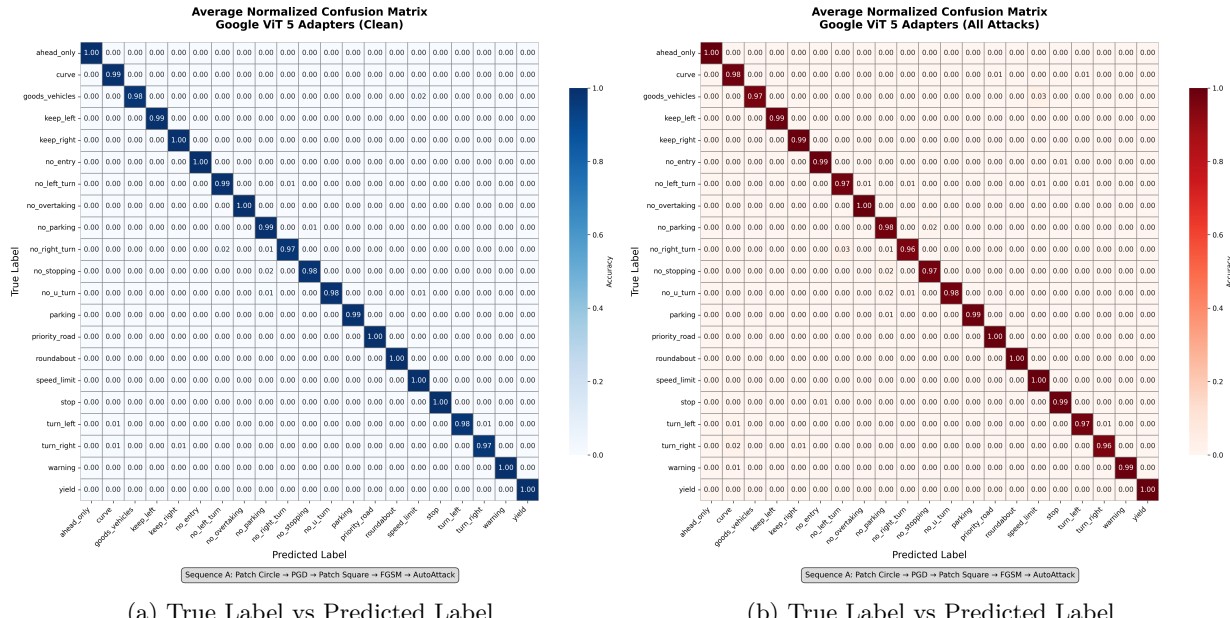

(a) True Label vs Predicted Label    (b) True Label vs Predicted Label

Figure 11: Google ViT with sequentially trained adapters (Sequence A): Confusion matrices comparing model performance on (a) clean and (b) adversarial test sets. Adversarial and clean test results are the aggregate results of each quintuple adapter merged model. Adversarial test results are the aggregate results of each attack.

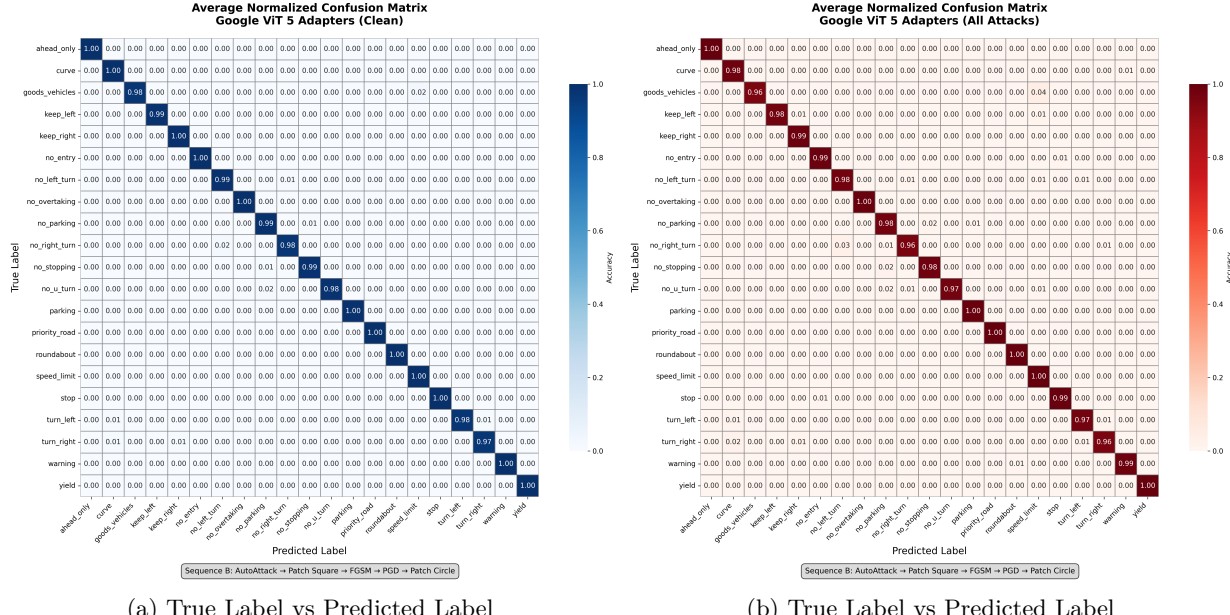

(a) True Label vs Predicted Label       (b) True Label vs Predicted Label

Figure 12: Google ViT with sequentially trained adapters (Sequence B): Confusion matrices comparing model performance on (a) clean and (b) adversarial test sets. Adversarial and clean test results are the aggregate results of each quintuple adapter merged model. Adversarial test results are the aggregate results of each attack.

### B.0.2   Swin-B parameter averaged adapters

Parameter-averaged LoRA adapters for Swin-B (Figure 13 - Figure 17) exhibit the same behaviors and flaws illustrated by the parameter averaged adapters in Section 5. The single adapter model performs as designed with average accuracy on both adversarial and clean test data (Figure 13). However, as additional adapters are composed, the model starts to experience interference in the parameter space, causing confusion between classes (Figure 15), and eventually leading to failure and catastrophic forgetting (Figure 17).

### B.0.3   Swin-B sequentially trained adapters

The results of sequential training illustrate the same two attack sequences defined in Section 5.3.2 and can be seen in Figure 18 and Figure 19.

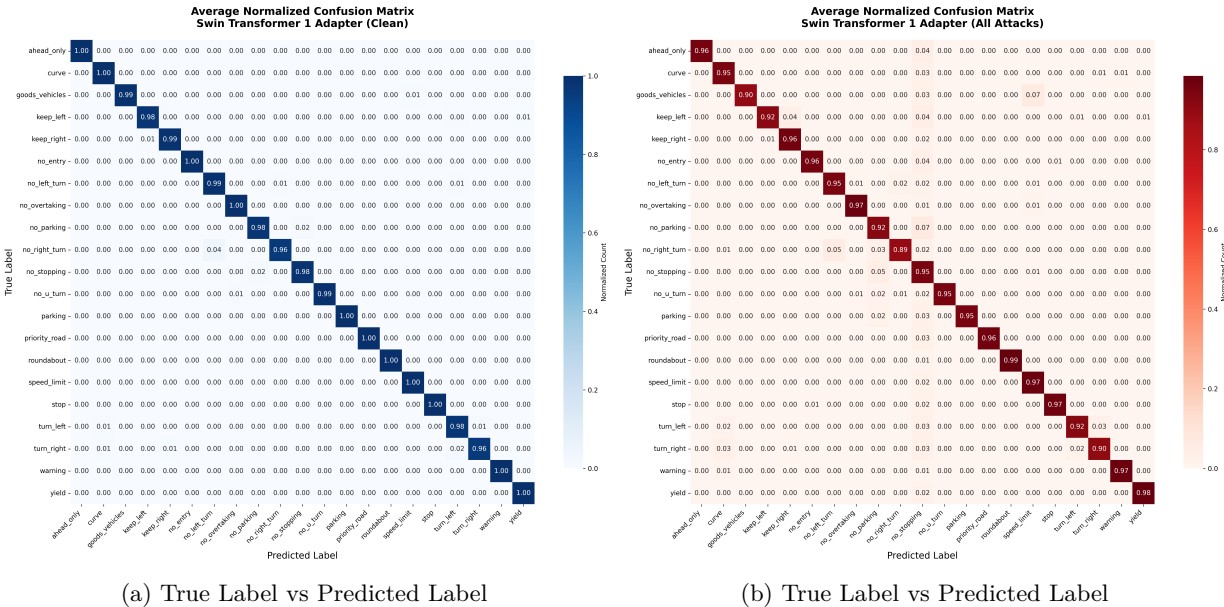

(a) True Label vs Predicted Label

(b) True Label vs Predicted Label

Figure 13: Swin-B with one parameter-averaged adapter performance: Confusion matrices comparing model performance on (a) clean and (b) adversarial test sets.

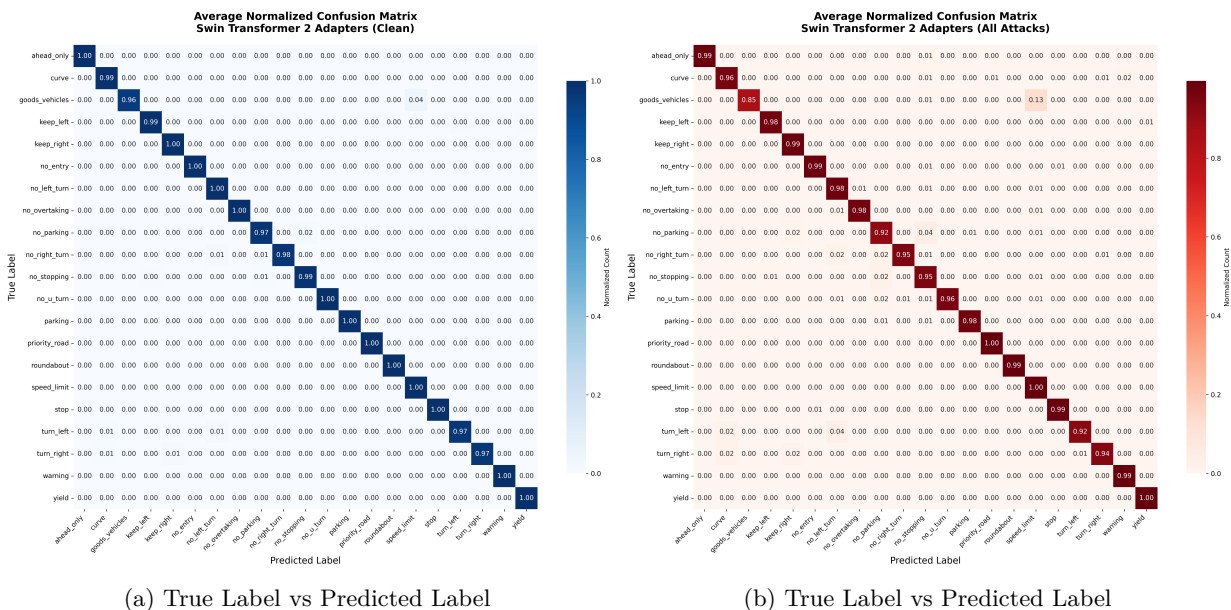

(a) True Label vs Predicted Label

(b) True Label vs Predicted Label

Figure 14: Swin-B with two parameter-averaged adapters performance: Confusion matrices comparing model performance on (a) clean and (b) adversarial test sets.

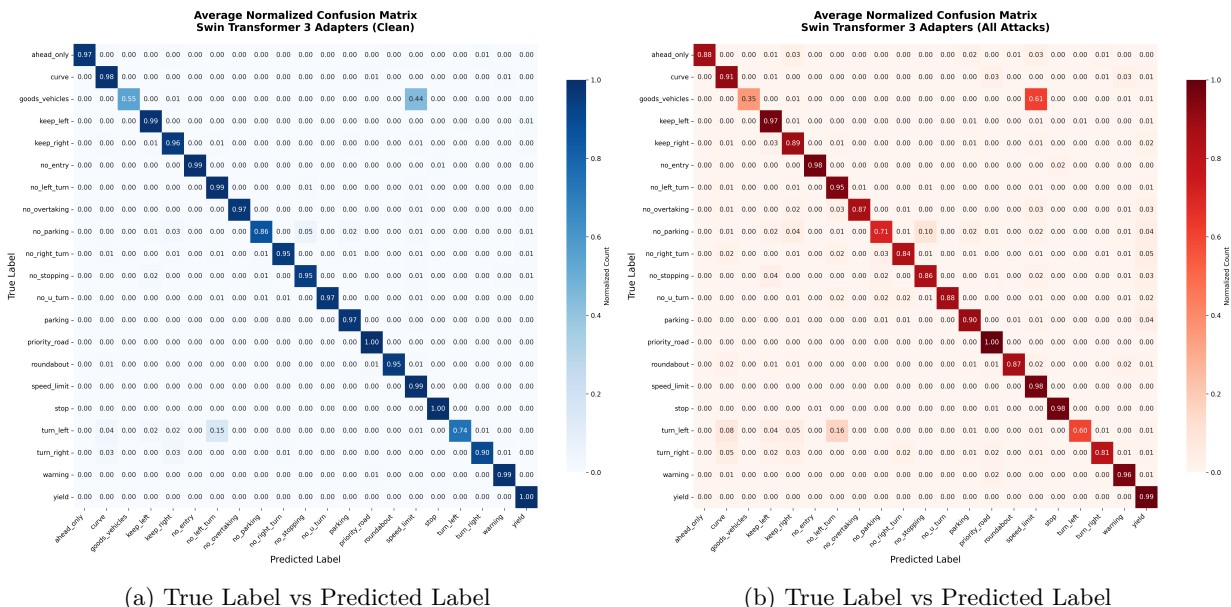

(a) True Label vs Predicted Label      (b) True Label vs Predicted Label

Figure 15: Swin-B with three parameter-averaged adapters performance: Confusion matrices comparing model performance on (a) clean and (b) adversarial test sets.

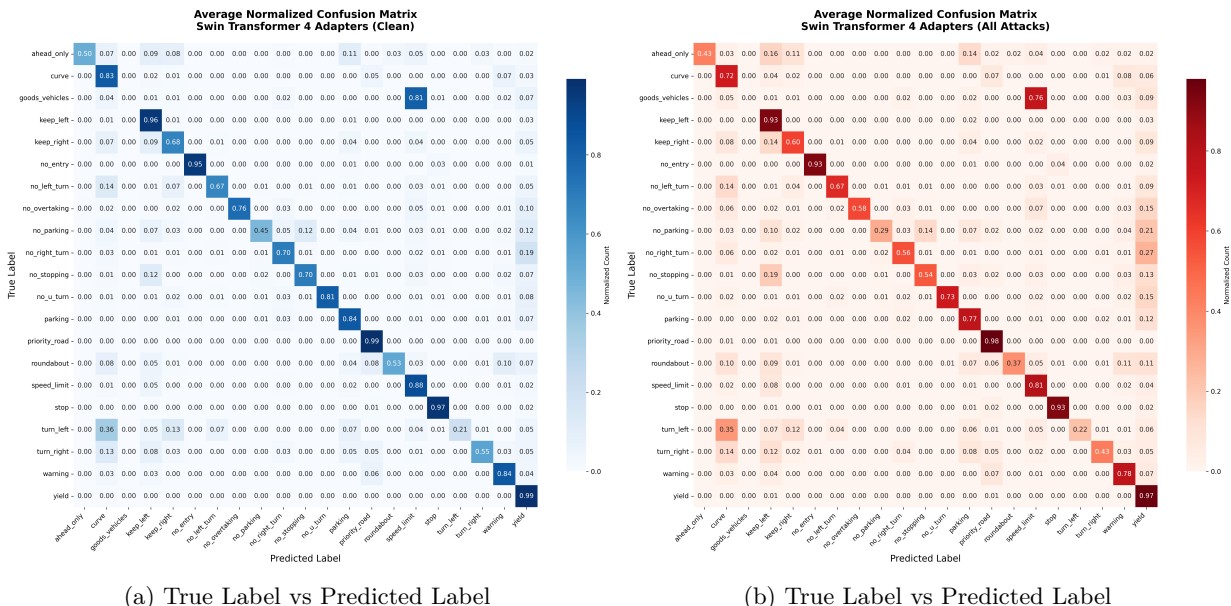

(a) True Label vs Predicted Label      (b) True Label vs Predicted Label

Figure 16: Swin Transformer with four parameter-averaged adapters performance: Confusion matrices comparing model performance on (a) clean and (b) adversarial test sets.

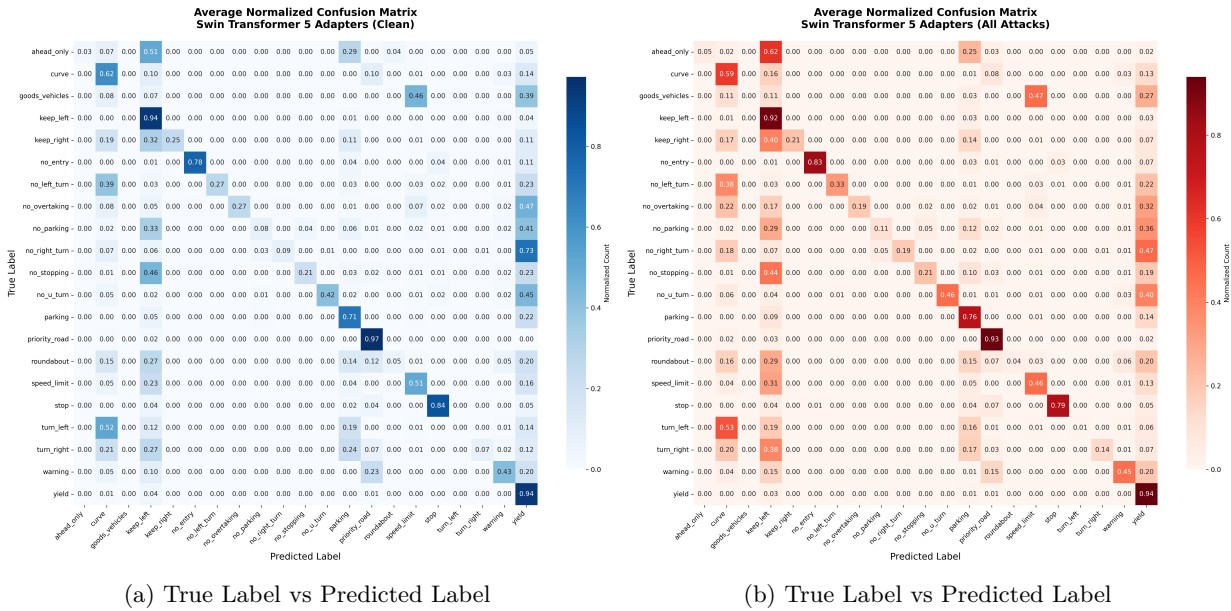

(a) True Label vs Predicted Label      (b) True Label vs Predicted Label

Figure 17: Swin-B with five parameter-averaged adapters performance: Confusion matrices comparing model performance on (a) clean and (b) adversarial test sets.

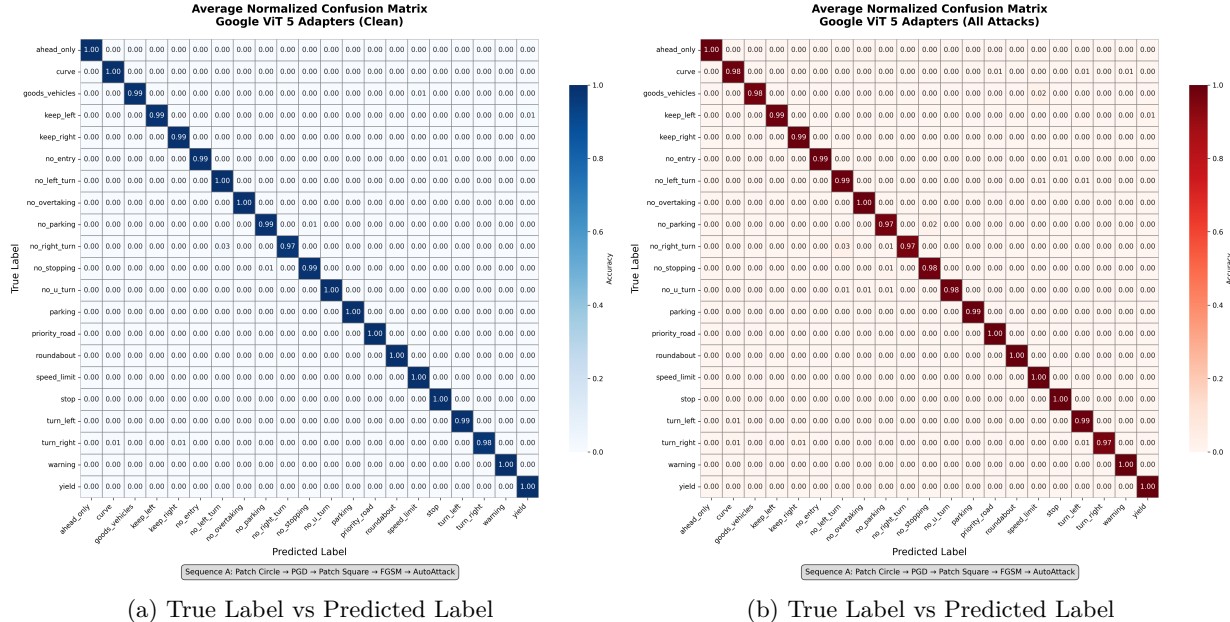

(a) True Label vs Predicted Label      (b) True Label vs Predicted Label

Figure 18: Swin-B with sequentially trained adapters (Sequence A): Confusion matrices comparing model performance on (a) clean and (b) adversarial test sets.

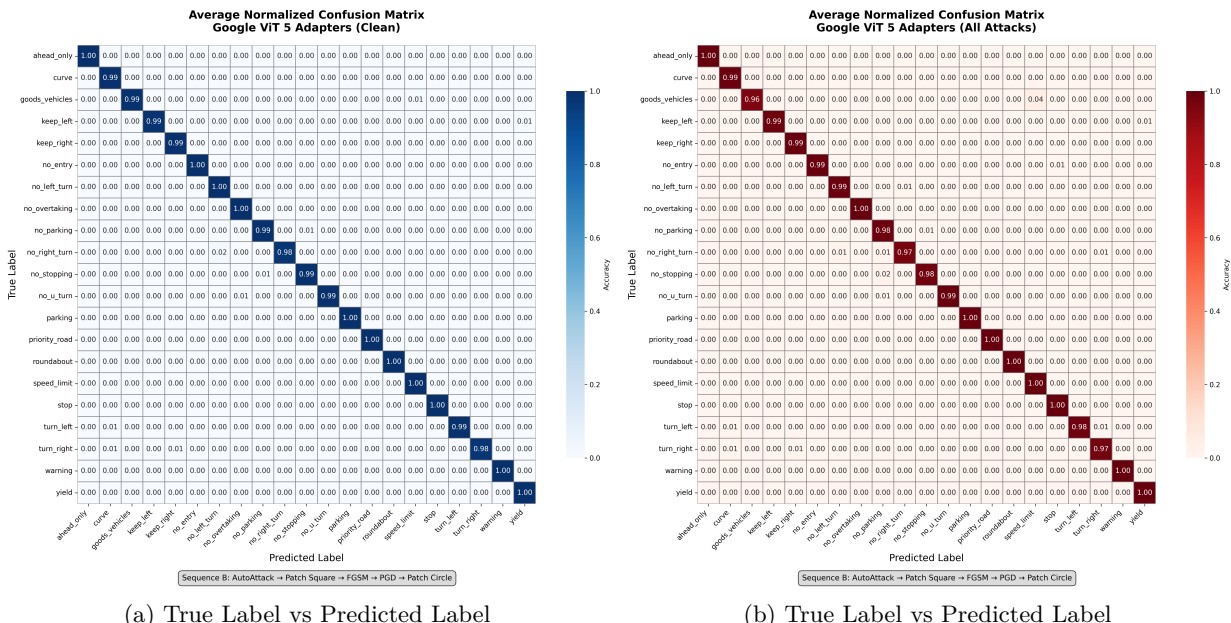

(a) True Label vs Predicted Label        (b) True Label vs Predicted Label

Figure 19: Swin-B with sequentially trained adapters (Sequence B): Confusion matrices comparing model performance on (a) clean and (b) adversarial test sets.

