# OpenReview forum: "Elytra: A Flexible Framework for Securing Large Vision Systems"
_TMLR — Withdrawn by Authors_

### Review · Reviewer_YbpR · 2026-03-23

**Summary Of Contributions:**

This paper investigates the application of Low-Rank Adaptation (LoRA) as a mechanism for applying "security patches" to Vision Transformers (ViT) and Swin Transformers. The authors focus on hardening these models against common adversarial attacks like PGD, FGSM, and AutoAttack in the context of traffic sign classification. The primary empirical observation is that combining multiple LoRA-based patches via simple parameter averaging leads to performance degradation , whereas sequential training allows the model to accumulate defenses.

Key Strengths:
- The paper addresses a practical concern regarding the computational cost of adversarial training for large-scale models.

- The inclusion of AutoAttack as a baseline ensures that the reported robustness is not merely a result of gradient masking.

Key Weaknesses:
- The core methodology is a direct application of LoRA. The concept of "patching" is essentially just task-specific fine-tuning using PEFT, which is already a standard practice in the field.

- The conclusion that sequential training outperforms naive averaging is well-known in the literature regarding continual learning and LoRA composition. The paper does not offer a novel theoretical or algorithmic solution to the interference problem, but rather adopts a standard sequential training approach.

- Despite the title's claim of a "Flexible Framework for Large Vision Systems" , the experiments are confined to a very specific image classification task on traffic signs.

**Audience:**

No

**Audience Explanation:**

The current findings are likely too incremental for the TMLR audience. The application of LoRA to fine-tune models is now a routine technique. The main discovery that sequential training avoids parameter interference is a foreseeable result that has been discussed in the context of Model Merging and Continual Learning. Without a more significant technical contribution or a much broader evaluation across different vision tasks, the paper provides little new information to the machine learning community.

**Broader Impact Concerns:**

The authors acknowledge the potential dual-use of their work, specifically that the methodology could be repurposed to efficiently inject backdoors into large models. This is a valid concern. However, as it is a common issue with most fine-tuning and security-related research, the current Broader Impact Statement is sufficient.

**Claims And Evidence:**

Yes

**Claims Explanation:**

The authors provide extensive empirical evidence using two major Vision Transformer architectures across a variety of adversarial attacks. The data clearly supports the claim that sequential LoRA training can restore accuracy on adversarial samples (improving it by up to 24.09%) without significantly compromising clean accuracy. The comparison between parallel and sequential training (Figures 4 and Tables 5/6) is particularly convincing in demonstrating the limitations of simple parameter averaging.

While the numerical results support the effectiveness of sequential training for the specific task provided , the evidence for ELYTRA being a "flexible framework for large vision systems"  is insufficient. A "flexible framework" for vision systems should demonstrate efficacy beyond simple 21-class classification. Modern vision systems in autonomous driving rely on object detection and segmentation, neither of which are addressed.Furthermore, the comparison is limited to "Adversarial Hardening". Without comparing ELYTRA to other state-of-the-art Parameter-Efficient Fine-Tuning or Adversarial Defense methods, it is unclear if the observed benefits are unique to the proposed approach or simply a baseline property of low-rank fine-tuning.

**Requested Changes:**

- To justify the "Flexible Framework" claim, the authors must demonstrate the approach on a more complex task than classification, such as Adversarial Object Detection on a standard benchmark (e.g., COCO or specialized autonomous driving datasets).

- Compare ELYTRA against other PEFT methods (e.g., Adapters, BitFit, or Prefix Tuning) to isolate the impact of the "low-rank" design choice.

- Since sequential training is the proposed solution to interference , the authors should compare their method with existing continual learning baselines (e.g., EWC or Replay-based methods) that are designed to mitigate catastrophic forgetting.

- Provide a rigorous analysis of the parameter space to explain why the sequential update in Equation 6 results in a stable point, whereas the parallel update in Equation 4  does not.

- Perform an ablation study on the impact of attack order in the sequential training of 5+ adapters to see if the model eventually saturates or degrades.

---

### Review · Reviewer_L7Se · 2026-03-27

**Summary Of Contributions:**

This paper focuses on large vision models in autonomous driving, especially traffic sign classification models, which are vulnerable to adversarial attacks. For example, imperceptible perturbations or local patches/stickers can cause a model to misclassify a Stop sign. The authors argue that conventional defenses typically either require fine-tuning the entire model or rely on input preprocessing; the former is costly, while the latter often generalizes poorly. Motivated by this, they ask whether one can treat defense as a form of security patching, analogous to software systems, by attaching a lightweight "security adapter" to the vision model. Concretely, the paper proposes adding a small LoRA adapter to a pretrained large vision model as an on-demand "security patch". The authors call this framework Elytra.

Strengths

1. The paper reframes adversarial robustness as a patch management problem. This is a practically meaningful and potentially useful framing.

2. The paper addresses an important but often overlooked issue: the combination of multiple patches. Real systems face a sequence of different attacks rather than a single threat model. The authors explicitly compare the failure of parallel composition against the relative success of sequential composition, which makes the paper more deployment-relevant than work that only studies a single defense in isolation.

Weaknesses

1. The core idea is not sufficiently novel. There has already been substantial prior work on improving adversarial robustness by freezing a large vision model and training only a small number of additional parameters, such as adapters, prompts, LoRA modules, or SVD-based parameters. For example, [1] also performs adversarial fine-tuning with LoRA on top of a frozen pretrained ViT to improve robustness; [2] studies how to combine multiple defense-specific LoRAs; [3] proposes LoRA-based parameter-efficient adversarial adaptation with additional designs such as clustering, alignment, and adaptive updates; and [4][5][6] are also closely related. In my view, Revisiting Adapters (2022), FullLoRA (2024), ADAPT (2025), and AdvLoRA (2024) already cover much of the space explored by this submission.

2. The title refers to "large vision systems", but the experiments are in fact concentrated on traffic sign classification, rather than detection, segmentation, or multimodal perception. This is still far from a realistic autonomous driving system. The paper does not evaluate an end-to-end perception stack, a detection model, or a multitask driving system; instead, it studies a heavily simplified cropped classification problem. In essence, this is only a digital attack study on cropped traffic sign classification.

3. The paper does not provide sufficiently convincing real-world physical attack validation. The data do not cover realistic physical instantiations of digital attacks, and domain transfer to real-world settings is not verified.

4. Figure 1 explicitly presents a threat landscape including C&W attack, Physical Patch, and Future Attack N, but the actual experiments only evaluate FGSM, PGD, Patch Attack, and AutoAttack. There are no results for C&W, and there is no genuine physical-world evaluation.

5. The experimental definition is unclear. In the main text, the paper says that LoRA is applied to multi-head attention blocks, MLPs, and the classification head. However, Appendix A.3 states that MLP blocks and key projections are not modified, and that only the query, value, and classification layer are updated. Then Table 9 lists the target modules as query, key, value, output.dense. These descriptions are inconsistent.

6. The baseline numbers are also inconsistent. In Tables 2/3/4/5, the ViT PGD baseline is 75.14, but in Table 6 it becomes 77.61, with no explanation provided.

7. The experimental results lack statistical support. There are no standard deviations, no confidence intervals, no repeated runs with multiple random seeds, and no significance tests.

8. The baselines are insufficient. The paper cites several prior works on LoRA composition / orthogonalization, yet does not compare directly against these methods. It also does not compare against other PEFT baselines, a single LoRA jointly trained on multiple attacks, or stronger adapter merging approaches.

9. It is unclear what the actual advantage of LoRA is over much simpler alternatives, such as fine-tuning only the output layer or training only the classification head.

10. Beyond the omission of relevant LoRA-composition baselines, the paper also overlooks a substantial body of lightweight adversarial defense methods more broadly.

[1] FullLoRA: Efficiently Boosting the Robustness of Pretrained Vision Transformers

[2] Hyper Adversarial Tuning for Boosting Adversarial Robustness of Pretrained Large Vision Models

[3] Enhancing Adversarial Robustness of Vision-Language Models through Low-Rank Adaptation

[4] ADAPT to Robustify Prompt Tuning Vision Transformers

[5] Few-Shot Adversarial Low-Rank Fine-Tuning of Vision-Language Models

[6] Revisiting Adapters with Adversarial Training

**Audience:**

Yes

**Audience Explanation:**

The framing of adversarial robustness as patch management is interesting, and the discussion of composing lightweight defense modules may be relevant to researchers working on robust and parameter-efficient adaptation.

**Claims And Evidence:**

No

**Claims Explanation:**

The claims are overstated relative to the evidence. The experiments are limited in scope, the paper contains inconsistencies in experimental details and baseline numbers, and the results lack statistical support.

**Requested Changes:**

The paper should substantially narrow its claims to match the actual experimental scope and avoid overstating its relevance to large vision systems or autonomous driving. It also needs to resolve the inconsistencies in experimental setup and reported baseline numbers, add stronger and more appropriate baselines, and provide proper statistical support for the results. Finally, the authors should clarify the specific advantage of LoRA over simpler lightweight alternatives and better position the work within the existing literature on parameter-efficient adversarial defense.

---

### Review · Reviewer_ijBa · 2026-04-08

**Summary Of Contributions:**

This work introduces Elytra, a framework that reframes adversarial robustness for large vision models as a post-hoc patching problem rather than full retraining. The key is to learn lightweight, attack-specific LoRAs that can be attached to a frozen backbone.

Instead of updating all model parameters, Elytra trains small low-rank updates to defend against specific adversarial threats. A central contribution is the observation that while naïvely combining multiple such adapters leads to severe interference, sequentially training and composing them enables effective multi-threat robustness. Empirically, the method shows substantial robustness improvements across several attacks while maintaining clean accuracy and reducing trainable parameters by orders of magnitude.

The main strength is that treating robustness as modular security patching is compelling for real-world deployment. Based on this paper, this approach is also simple but effective, which  use LoRA techniques to achieve strong empirical gains with small computational cost. The idea about sequential training seems practical in real applications. This paper discusses different considerations in patch combinations, which show the engineering value of this work. The paper is well written and clear.

As for drawbacks, it primarily repurposes existing LoRA techniques.  This work lacks a theoretical explanation or  at least more rigorous analysis for why sequential composition works, given that autonomous driving systems need trust about its reliability.
As mentioned in this paper, the evaluation is relatively narrow (focused on traffic sign classification and a small set of transformer models), and robustness is mostly discussed in digital attack settings.

**Audience:**

Yes

**Audience Explanation:**

The paper addresses a practically relevant problem how to incrementally improve robustness of already-deployed models without full retraining and proposes a simple, modular solution. The idea of treating robustness as composable security patches is conceptually appealing and may resonate with researchers interested in model maintenance and system-level ML security.

**Claims And Evidence:**

Yes

**Claims Explanation:**

The evaluation covers multiple strong attacks like PGD, AutoAttack, patch attacks etc., and two different vision transformer architectures, and results consistently show substantial robustness gains with minimal impact on clean accuracy. The comparison against full adversarial fine-tuning also supports the claim of parameter efficiency. The evidence is clear and convincing for one specific experimental setting considered (traffic sign classification). But it seems not sufficient enough to fully substantiate the generality of the claims, such as the experiments are limited to a single task and a narrow set of models

**Requested Changes:**

See drawbacks mentioned before

---

### Note · Authors · 2026-05-05

I have read and agree with the venue's withdrawal policy on behalf of myself and my co-authors.